

# Seasonality of aerosol optical properties in the Arctic

Lauren Schmeisser[1,3*], John Backman[2], John A. Ogren[1,3], Elisabeth Andrews[1], Eija Asmi[2], Sandra Starkweather[1,3], Taneil Uttal[3], Markus Fiebig[4], Sangeeta Sharma[5], Kostas Eleftheriadis[6], Stergios Vratolis[6], Michael Bergin[7], Peter Tunved[8], Anne Jefferson[1]

[1]University of Colorado, Cooperative Institute for Research in Environmental Sciences, Boulder, CO, USA
[2]Finnish Meteorological Institute, Atmospheric Composition Research, Helsinki, Finland
[3]National Oceanic and Atmospheric Administration, Earth System Research Laboratory, Boulder, CO, USA
[4]Norwegian Institute for Air Research, Kjeller, Norway
[5]Environment and Climate Change Canada, Science & Technology Branch, Climate Research Division, Toronto, Canada
[6]Institute of Nuclear and Radiological Science & Technology, Energy & Safety, Environmental Radioactivity Laboratory, NCSR Demokritos, Athens, Greece
[7]Duke University, Department of Civil & Environmental Engineering, Durham, NC, USA
[8]Stockholm University, Department of Environmental Science and Analytical Chemistry, Stockholm, Sweden
*Now at University of Washington, Department of Atmospheric Sciences, Seattle, WA, USA

*Correspondence to*: Lauren Schmeisser (lauren.schmeisser@gmail.com)

**Abstract**

Given the sensitivity of the Arctic climate to short-lived climate forcers, long-term in-situ surface measurements of aerosol parameters are useful in gaining insight into the magnitude and variability of these climate forcings. Seasonality of aerosol optical properties, including aerosol light scattering coefficient, absorption coefficient, single scattering albedo, scattering Ångström exponent, and asymmetry parameter are presented for six monitoring sites throughout the Arctic: Alert, Canada; Barrow, USA; Pallas, Finland; Summit, Greenland; Tiksi, Russia; and Zeppelin Mountain, Ny-Ålesund, Svalbard, Norway. Results show annual variability in all parameters, though the seasonality of each aerosol optical property varies from site to site. There is a large diversity in magnitude and variability of scattering coefficient at all sites, reflecting differences in aerosol source, transport and removal at different locations throughout the Arctic. Of the Arctic sites, the highest annual mean scattering coefficient is measured at Tiksi (12.47 Mm$^{-1}$) and the lowest annual mean scattering coefficient is measured at Summit (1.74 Mm$^{-1}$). At most sites, aerosol absorption peaks in the winter and spring, and has a minimum throughout the Arctic in the summer, indicative of the Arctic haze phenomenon; however, nuanced variations in seasonalities suggest that this phenomenon is not identically observed in all regions of the Arctic. The highest annual mean absorption coefficient is measured at Pallas (0.48 Mm$^{-1}$) and Summit has the lowest annual mean absorption coefficient (0.12 Mm$^{-1}$). At the Arctic monitoring stations analyzed here, mean annual single scattering albedo ranges from 0.909-0.960 (at Pallas and Barrow, respectively), mean annual scattering Ångström exponent ranges from 1.04-1.80 (at Barrow and Summit, respectively), and mean asymmetry parameter ranges from 0.57-0.75 (at Alert and Summit, respectively). Systematic variability of aerosol optical properties in the Arctic supports the notion that the sites presented here measure a variety of aerosol populations, which also experience different removal mechanisms. A robust conclusion from the seasonal cycles presented is that the Arctic cannot be treated as one common and uniform environment, but



rather is a region with ample spatio-temporal variability in aerosols. This notion is important in considering the design or aerosol monitoring networks in the region, and is important for informing climate models to better represent short-lived aerosol climate forcers in order to yield more accurate climate predictions for the Arctic.

## 1. Introduction

The Arctic is a unique environment, characterized by sensitive interactions and feedbacks between the atmosphere, ocean, cryosphere and biosphere (Serreze and Francis, 2006; Serreze and Barry, 2011). In recent decades, substantial changes have been observed in the Arctic, including increases in air temperature (Johannessen et al., 2004), decreases in sea ice extent and thickness (Lindsay and Zhang, 2005; Stroeve et al., 2007; Stroeve et al., 2012), changes in Arctic vegetation (Wang and Overland, 2004; Chapin et al., 2005; Pearson et al., 2013), and shifts

in precipitation patterns (Groves and Francis, 2002; Bintanja et al., 2014). The mechanisms behind these changes are induced by anthropogenic global climate change (Anisimov et al., 2007), and have not yet been fully characterized. Human presence, and thus emissions, in the Arctic are likely to increase in the future due to decreases in sea ice making the region more accessible for energy extraction and shipping activities (e.g., Aliabadi et al., 2015; Eckhardt et al., 2013). More research in the Arctic, particularly on atmospheric components and processes in the region, is

necessary to better understand what is changing, why it is changing, and how it might change in the future (Anisimov et al., 2007).

    Within the Arctic atmosphere, short-lived climate forcers like aerosols are important contributors to the observed warming and environmental changes in the region (Quinn et al., 2008; Najafi et al., 2015). Aerosols can affect the climate both directly by scattering and absorbing incoming solar radiation, and indirectly through aerosol-cloud

interactions (Twomey, 1977). Quantifying the forcing by aerosols in the Arctic is especially complex, given the annual variability in surface albedo and cloudiness, the stratified atmosphere, resulting feedbacks, and long-range aerosol transport. Measurements of surface Arctic aerosol optical properties in particular can help define and constrain inter-annual, seasonal and diurnal variability of light scattering and absorption, potential particle sources, and resulting radiative forcing. The observation capacity demonstrated here has potential for providing in situ

observational checks on long-term black carbon inventories and monitoring strategies of importance to international pollution mitigation effects. This paper will seek to provide an overview of surface aerosol optical properties in the Arctic.

## 2. Background

    Observations of aerosols in the Arctic have a long (>50 yr) history (e.g., Mitchell's (1957) report on so-called Arctic

haze layers), although continuous surface measurements of aerosol optical properties did not begin until the mid-1970s at Barrow, Alaska (BRW) and later at other sites. The start of long-term, continuous surface measurements, on-going to this day, have provided information about aerosol chemistry, microphysics and optical properties and enabled the development of aerosol climatologies, the analysis of trends and the evaluation of models. Such analyses have been driven by the need to understand the remote and local sources, transport and processes that influence





aerosol properties in the Arctic. Understanding aerosol optical properties in particular is important in gaining insight into the role of aerosols on the Arctic's radiative energy budget (e.g., Quinn et al., 2011).

Despite the challenges associated with performing high-quality, long-term atmospheric observations in the Arctic (e.g., high costs, extreme conditions, difficult access, etc.), several monitoring stations do currently exist in the

Arctic. Of these monitoring sites, 10 contribute to the International Arctic Systems for Observing the Atmosphere (IASOA) network. The purpose of the IASOA organization is twofold: (1) To enhance interoperable observational abilities and coverage of surface atmospheric monitoring in the data-sparse Arctic, and (2) To foster pan-Arctic scientific collaboration with easier data access and strengthened synergy among researchers (Uttal et al., 2016). Of the 10 monitoring sites, 6 stations have multi-year, continuous measurements of aerosol optical properties, and it is

these data from years 2012-2014 that are used for the Arctic aerosol analysis presented in this paper. These monitoring stations follow standardized aerosol sampling protocol, as advised by the Global Atmosphere Watch (GAW) network (http://library.wmo.int/opac/index.php?lvl=notice_display&id=19622), and contribute to a coordinated data archive (i.e., the World Data Center for Aerosols (WDCA) hosted at the Norwegian Institute for Air Research (http://ebas.nilu.no/)).

Published climatologies and seasonality of surface extensive aerosol optical properties (i.e., properties that depend on the amount of aerosol) have shown that at many Arctic sites, scattering and absorption are highest in the late winter and early spring, and lowest in the summer (e.g., Bodhaine, 1983 (Barrow); Bodhaine, 1995 (Barrow); Sharma et al., 2004 (Alert); Eleftheriadis et al., 2009 (Zeppelin); Heintzenberg, 1982 (Zeppelin); Aaltonen et al., 2006 (Pallas); Lihavainen et al., 2015 (Pallas)). However, results shown here will support the notion that not all

Arctic sites have this seasonal cycle. The winter/spring aerosol enhancement is called Arctic haze, referring back to Mitchell's (1957) early airborne observations. Understanding the sources, characteristics, and effects of Arctic haze has been a continuing effort over the past several decades (e.g., Rahn et al., 1977; Shaw, 1995; Quinn et al., 2007; Liu et al., 2015; and references therein). The low summertime values of absorption and scattering currently observed in the Arctic are likely to be particularly vulnerable to warmer, drier climatic conditions (e.g., due to increases in

summertime forest fires and decreases in sea ice leading to enhanced marine emissions and human activities in the region during the summer). Published climatologies and seasonal cycles of in-situ Arctic aerosol intensive properties (i.e., properties that are ratios of extensive properties and not directly dependent on aerosol amount) are sparse and suggest that, unlike the relatively consistent seasonal pattern for extensive properties, the seasonal cycles of intensive optical properties (e.g., Ångström exponent) may differ from site to site (Delene and Ogren, 2002;

Aaltonen et al., 2006, Lihavainen et al., 2015). This work seeks to expand on previous aerosol optical analyses in the Arctic by synthesizing aerosol seasonality at multiple Arctic stations, and adding new knowledge on the seasonality of intensive aerosol characteristics in the region.

At present, only surface measurements can provide a seasonal context for the range of aerosol optical properties used to determine radiative forcing efficiency (RFE), including absorption, scattering, backscattering fraction,

asymmetry parameter and single scattering albedo. While vertical profiles are important due to the stratified



conditions in the Arctic atmosphere (e.g., Rahn et al., 1977) aircraft campaigns in the Arctic thus far do not provide insight into seasonality. Stone et al. (2014; their Fig. 5) notes that only one aircraft campaign in the last 30 years occurred outside the Arctic Haze period. Remote sensing instruments such as sun photometers are limited due to long periods of darkness during the winter, and satellite measurements have limited utility due to the high albedo of

the Arctic snow surface and the dark Arctic winters. An additional limitation of remote sensing measurements is that parameters important for RFE calculations (e.g., single scattering albedo) cannot be retrieved without high uncertainties in the Arctic due to the low aerosol optical depth (AOD) (Dubovik et al., 2000). Although geographically sparse compared to the potential of remote sensing and aircraft campaigns, surface measurements have the advantage of being long-term, year-round and comprehensive.

The objective of this paper is to explore the seasonality and spatio-temporal variability of surface aerosol optical properties in the Arctic; the results of this exploration may be useful for continued improvement of modeling and remote sensing capabilities. Here we ask how aerosol optical properties differ among six Arctic monitoring sites; how monthly variability in aerosol optical properties compares across the sites; what systematic variability among aerosol optical properties exists in the Arctic; what pairing of trajectory data with aerosol optical properties suggests

about aerosol sources in the Arctic, and how this trajectory analysis varies geographically from station to station?

### 3. Methods

#### 3.1 Monitoring Sites

The analysis presented here uses in-situ measured aerosol properties from 6 Arctic monitoring stations. To be included in the analysis, a station had to have continuous and concurrent aerosol light scattering and two sets of

absorption measurements: (i) Aethalometer and (ii) 'reference' co-located absorption instrument (details in Sect. 3.2 Data and Instrumentation) during years 2012-2014. Six monitoring sites met these criteria: Alert, Canada (ALT); Barrow, Alaska (BRW); Pallas, Finland (PAL); Summit, Greenland (SUM); Tiksi, Russia (TIK); and Zeppelin Mountain, Ny-Ålesund, Svalbard, Norway (ZEP) (for a record of data availability at all IASOA sites, see the IASOA data access portal https://www.esrl.noaa.gov/psd/iasoa/dataataglance). The following sections describe the

location of, conditions at, and instrumentation at the sites analyzed here. Arctic stations not included in this study either do not measure the parameters presented here, or do not have continuous measurements for the period of interest. This time period was chosen to align with Backman et al. (2017), which presents an Arctic-specific correction scheme for aethalometer data, to be used here to describe absorption coefficients at each of the stations. More information on this correction scheme is presented in Section 3.2. Table 1 provides further information on

monitoring station location, instrumentation, and sampling inlet configuration. Figure 1 shows a map of the Arctic sites, as well as photos of the monitoring stations and their surroundings.

#### 3.1.1 Alert, Canada (ALT)




Alert is located in Nunavut, Canada and is operated by Environment and Climate Change Canada (ECCC). The aerosol optical property measurements are made in collaboration with the National Oceanic and Atmospheric Administration (NOAA). The monitoring station is the most northerly site in the GAW network, and despite the site being shared with a Canadian military facility and an ECCC upper air weather station, it is remote and far from

industrial pollution sources. The measurement laboratory was established in 1986, and has long-term Aethalometer measurements since 1989, and aerosol absorption (PSAP) and scattering measurements from 2005 on. The aerosol instruments measure from an inlet and aerosol system that has both 1 and 10 μm diameter size cuts, and data from the 10 μm size cut are used here. Relative humidity of the sample is consistently less than 40%, which is important in limiting effects of hygroscopic growth on the aerosol measurements. Instrument descriptions can be found in

Table 1. Previous work on aerosol optical properties at ALT can be found in Hopper et al. (1994), Sharma et al. (2002), Sharma et al. (2004), Sharma et al. (2006), and Quinn et al. (2007).

### 3.1.2 Barrow, Alaska (BRW)

The Barrow observatory was established in 1973 and is operated by NOAA with additional support from the U.S. Department of Energy and the National Science Foundation (NSF). The site is situated 5 km northeast of the town of

Barrow, Alaska (population ~4,200), and is 2 km from the Arctic Ocean coast. The station primarily measures regionally representative air masses coming off of the Beaufort Sea. Air masses coming from the direction of the town are marked as contaminated and those data are not used here. Aerosols are sampled through an inlet and aerosol system with a switching impactor that has both 1 and 10 μm size cuts, though only data from the 10 μm size cut are analyzed here. The Aethalometer samples air from a separate inlet with no aerosol size cut, and thus

measures the full aerosol size range. Previous descriptions of the aerosol optical property climatology from the older generation of instrumentation at BRW (see Table 1) are found in Bodhaine (1983), Bodhaine (1995), Delene and Ogren (2002), and Quinn et al. (2007).

### 3.1.3 Pallas, Finland (PAL)

The Pallas Atmosphere-Ecosystem Supersite is operated by the Finnish Meteorological Institute (FMI), and is a part

of the larger Pallas-Sodankylä GAW station located in northern Finland. The Pallas main research site is located in the Pallas-Yllästunturi National Park on the top of the Sammaltunturi fell at an elevation of 565 m asl and above the tree line. The nearest town is Muonio, located 19 km to the west with ~2500 inhabitants, though the station typically measures clean Arctic air masses due to a prevailing wind direction not affected by town contamination. The surrounding region is hilly and vegetated with pine, spruce, birch and low growing shrubs. The total aerosol inlet at

PAL is slightly heated to avoid freezing and to maintain RH below 40%. The Aethalometer is connected to the total aerosol inlet. The other optical measurements (MAAP and nephelometer) are connected to a 2.5 μm size cut inlet. A more detailed description of aerosol optical measurements and sampling can be found in Lihavainen et al. (2015) and in Backman et al. (2017). A climatology of aerosol optical properties at PAL is presented by Aaltonen et al. (2006) and Lohila et al. (2015).



### 3.1.4 Summit, Greenland (SUM)

The Summit monitoring station is located in Greenland, Denmark, and is supported and operated by Duke University in collaboration with NOAA Earth Systems Research Laboratory with financial aid from the NSF. The scattering and co-located absorption measurements at SUM were initiated in 2011 as part of a NOAA collaboration with Georgia Institute of Technology. Summit is unique from the other stations in this study due to its high elevation of 3238 m asl, meaning it often measures free tropospheric air. The station is very remote and has no nearby anthropogenic aerosol sources apart from scientific operations near the site; when air masses blow from the direction of the scientific camp, data are marked as contaminated and are not included in this analysis. The inlet at Summit has a 2.5 μm size cut, and samples have RH < 40%, since the temperature inside the instruments is much warmer than the temperature outside. VanCuren et al. (2012) has some description of past aerosol measurements made at SUM.

### 3.1.5 Tiksi, Russia (TIK)

The Tiksi Hydrometeorological Observatory in Yakutsk, Russia was formed through a collaboration between the Russian Federal Services for Hydrometeorological and Environmental Monitoring (Roshydromet), NOAA, FMI, and NSF. Though there has been a meteorological observatory at this location since the 1930s, the new international site was established in 2009. The site is located in northern Siberia in the Sakha Republic of Russia, just 500 m from the coast of the Laptev Sea and ~5 km outside of the town of Tiksi (population 4600). Air masses coming from the direction of the town are marked as contaminated and are not included in this analysis. The monitoring station is surrounded by a tundra landscape, as seen in the photo of the Tiksi monitoring site in Fig. 1. Air is sampled through a heated inlet that prevents ice buildup and minimizes hygroscopic effects on the measurements by keeping RH <40%, and has a 10 μm size cut. A detailed description of the Tiksi site can be found in Uttal et al. (2013) and a previous analysis of aerosols at TIK with a detailed description of the sampling system can be found in Asmi et al. (2016).

### 3.1.6 Zeppelin Mountain, Ny-Ålesund, Norway (ZEP)

The Zeppelin Mountain observatory is located on a small mountain 475 m asl, just south of the small research village of Ny-Ålesund (30-150 inhabitants, depending on time of year) on Svalbard island in Norway. The monitoring station is owned by the Norwegian Polar Institute and operated by the Norwegian Institute for Air Research (NILU), and the most recent version of the station building was constructed in the year 2000. The site is typically located above the inversion layer, and thus measures air masses with minimal contamination. Aerosol instruments sample from an inlet line that reaches room temperature (~21 °C) before measurement so that RH < 20%. The inlet line does not have a size cut. Past analyses of aerosol measurements at ZEP can be found in Ström et al. (2003), Stohl et al. (2006b) and Eleftheriadis et al. (2009).

### 3.2 Data and Instrumentation



Although monitoring networks offer scientists an opportunity for regional cross-station analyses of aerosol seasonality and climatologies, comparing data across monitoring sites requires caution. Care must be taken to ensure data are measured, edited, and corrected using comparable high-quality methods. Moreover, comparing the same aerosol property measured by different instrument types or models necessitates extra attention. This section
describes the data and steps taken to ensure comparability of those data for this analysis.

All 6 sites in this analysis have scattering measurements for years 2012-2014 from an integrating nephelometer (TSI model 3563) measuring at 3 wavelengths (450 nm, 550 nm, 700 nm). Corrections to the raw scattering coefficient measurements are necessary to account for light source and angular non-idealities, and the correction methods described in Anderson and Ogren (1998) were used to correct the scattering coefficient data presented here.

In this analysis, absorption data are available from Aethalometers as well as other, co-located filter-based absorption instruments (i.e., CLAP, PSAP, and/or MAAP) at each observatory. The Magee Aethalometers are the only common absorption instrument among the six stations presented here, and this paper synthesizes the absorption data from Aethalometers across the Arctic. The Aethalometer data are corrected using the new Arctic-specific Aethalometer correction scheme presented by Backman et al. (2017). We use the 'reference' co-located absorption instruments to
gauge whether the corrected Aethalometer data are similar to what is expected for absorption coefficient values from other absorption measurements at the stations. The different co-located absorption instruments and Aethalometer data are described below.

Co-located reference absorption data at ALT are from a 3-wavelength (467, 530, 660 nm) Radiance Research Particle Soot Absorption Photometer (PSAP-3W) and at ZEP are from a 1-wavelength (525 nm) custom built
Particle Soot Absorption Photometer (PSAP-1W). The PSAP collects aerosol particles on a filter, and relates the change in light transmission through the filter over time to the absorption coefficient of the deposited aerosol. PSAP data are corrected using the correction schemes from Bond et al. (1999) and Ogren (2010) to adjust for multiple scattering effects, filter loading, apparent absorption, flow bias, spot size bias, and spectral scattering. Correcting for apparent absorption requires concurrent measurements of aerosol light scattering, which are available from TSI
nephelometers at all 6 stations.

Co-located absorption data at BRW and SUM were measured using a Continuous Light Absorption Photometer (CLAP) at 3 wavelengths (467 nm, 528 nm, 652 nm). The CLAP is a NOAA designed and built instrument that is based on the PSAP design, except that it samples consecutively on eight filter spots on one large 47 mm filter, as opposed to the one spot available on the 10 mm PSAP filter. The CLAP's multi-spot functionality enables it to run
unattended for 8 times longer than the PSAP, making it ideal for remote, less frequently visited locations (Ogren et al., 2017). The CLAP data are corrected the same way as the PSAP using Bond et al. (1999) and Ogren (2010) corrections.

PAL and TIK co-located reference absorption data are from a Thermo Fisher Scientific Multi-Angle Absorption Photometer (MAAP) at 1 wavelength (637nm) (Müller et al., 2011). The MAAP is a filter-based absorption





instrument that measures filter transmittance as well as back-scattered light at two angles (Petzold and Schönlinner, 2004). The backscattering measurements at different angles allow the instrument to account for multiple scattering and apparent absorption effects. Due to the low concentrations in the Arctic, no post-processing corrections are needed (Hyvärinen et al., 2013).

5       In addition to the co-located absorption measurement, all monitoring stations have absorption data collected from some model of the Magee Aethalometer. During 2012-2014, five of the stations- ALT, BRW, ZEP, PAL, and TIK- operated a 7-wavelength (370, 470, 520, 590, 660, 880, and 950 nm) Aethalometer AE31, while SUM operated a 1-wavelength (880 nm) Aethalometer AE16. The Aethalometer measures light transmitted through a filter on which particles are deposited and interprets the change in transmittance, or the attenuation of light through the filter, as the

aerosol light absorption, which the instrument reports as an atmospheric concentration of equivalent black carbon (eBC) (Petzold et al., 2013) particles using a mass absorption cross section of black carbon. There are known artifacts associated with measuring absorption coefficients on the Aethalometer filter tape, including multiple scattering by filter fibers, scattering by aerosol deposited on the filter, and decrease in sensitivity with increased filter loading. Many Aethalometer correction schemes exist that try to account for one or all of these artifacts (e.g.,

Collaud Coen et al., 2010; Drinovec et al., 2015), including GAW recommendations for the AE31 contained in GAW report 227 (https://library.wmo.int/opac/doc_num.php?explnum_id=3073), but there is currently no agreed upon or widely accepted correction scheme. Here we use a new Arctic-specific Aethalometer correction factor from Backman et al. (2017) to derive the light absorption coefficient from the Aethalometer data.

Backman et al. (2017) present an Arctic-specific multiple scattering enhancement factor, $C_f$, derived from

Aethalometer data and co-located absorption data from the same sites and time period used in this study. For all wavelengths and for the five low-altitude sites (ALT, BRW, PAL, TIK, ZEP), the value for $C_f$ was found to be 3.20, with interquartile values of 2.72-3.84. The Arctic correction factor is used to correct Aethalometer data using Eq. (1):

$$C_f = \frac{\sigma_o}{\sigma_{ap}} \quad (1)$$

where $\sigma_o$ is the uncorrected Aethalometer absorption coefficient and $\sigma_{ap}$ is the actual absorption coefficient that is corrected for multiple scattering by the filter fibers. Note that this correction scheme does not consider scattering by particles deposited on the Aethalometer filter or sensitivity of measurements to Aethalometer filter loading.

The Aethalometer absorption data corrected with the Backman et al. (2017) correction factor are compared to absorption coefficients from the co-located absorption instruments to ensure that the corrected Aethalometer data are

similar to absorption coefficients that are measured by other absorption instruments at the site. Figure 2 shows a time series of monthly median corrected Aethalometer data and co-located absorption data from 2012-2014 at each site. Data are adjusted to a co-located absorption instrument wavelength, except for SUM data, where the co-located absorption data are adjusted to the wavelength of the 1-wavelength Aethalometer (880 nm). Wavelengths of the data




in Fig. 2 are: 467 nm at ALT, 467 nm at BRW, 637 nm at PAL, 880 nm at SUM, 637 nm at TIK, and 525 nm at ZEP. Note that the y-axis scales in Fig. 2 are different for each site. Additional scatter plots comparing Aethalometer and co-locatedabsorption data, including $R^2$ values, can be found in supplemental materials. In general, the corrected Aethalometer absorption coefficients compare well to the co-located absorption coefficients, though the

comparability differs with season and site. ALT and BRW show good agreement between both absorption coefficient datasets ($R^2$=0.809 for ALT, $R^2$=0.839 for BRW) throughout the entire time. At BRW, there is a small systematic bias such that the co-located absorption values are slightly higher than the corrected Aethalometer absorption values. PAL also shows good agreement ($R^2$= 0.779) between absorption measurement techniques for the given time, apart from January 2013, which does not compare as well as the other months. Review of the PAL data

revealed no immediately apparent problems that could explain the anomalous results in January 2013. SUM has the worst agreement between co-located absorption data and corrected Aethalometer absorption data ($R^2$= 0.384), with higher biases in the winter and spring, and better agreement in the datasets in the summer. SUM data were not used in the development of the Backman et al. (2017) Arctic-specific Aethalometer correction scheme, which could be a factor in the larger differences in absorption values at that site. Additionally, the exceptionally clean air measured at

SUM means the instruments may frequently be measuring below detection limit, which could impact instrument agreement. TIK Aethalometer data are available for the entire 2012-2014 period, but the co-located MAAP absorption data only begin in summer of 2013, which is seen in Fig. 2(e). Concurrent Aethalometer and MAAP absorption measurements from 2013-2014 at TIK agree very well ($R^2$=0.851). ZEP absorption datasets also generally agree on the data seasonality, though there appears to be some seasonal bias in the agreement, with the

best correlation in the summer and larger differences in the corrected Aethalometer and co-located absorption data in the winter, resulting in lower overall agreement between measurement techniques ($R^2$=0.364).

Although agreement between Aethalometer measured absorption and co-located instrument absorption is imperfect and variable among stations, corrected Aethalometer data from all sites are utilized in the remainder of this paper for analyses of absorption coefficients at all six Arctic monitoring stations. Using Aethalometer measurements at each

location, rather than three different types of co-located reference instruments (PSAP, CLAP, and MAAP), eliminates issues with comparing data from different measurement techniques across stations. Furthermore, despite the differences in instrument agreement highlighted above, much of the difference in Aethalometer and co-located reference absorption values fall within combined instrumental uncertainties, as discussed later in this section.

Measurements from all instruments used in the analysis are reported at standard temperature and pressure (STP, T=

0 °C and P=1013 hPa).  The measurements are made at low RH (RH < 40%) to eliminate the confounding effect of water uptake. It is not difficult to maintain a low sample RH at these sites, even for sites without heated inlets, because the ambient dewpoint temperature is usually much lower than the temperature in the heated laboratories.

Quality assurance and quality control procedures were applied to the datasets at all six stations. Station scientists looked at each week of data individually to determine validity of the measurements. Additionally, there was a

second stage of data review by the authors of this paper to double-check the data quality. During time periods where



instruments appeared to be malfunctioning, or data were obviously influenced by local pollution (i.e., not representative of regional aerosol), data were invalidated or marked as contaminated. This helps ensure that data included here are representative of regional Arctic aerosol. At the sites in the study, measurements of absorption and scattering are made sub-hourly (data frequency 1-5 minutes), though all data used in the analysis are hourly averages

to improve the signal to noise ratio at the clean Arctic locations.

The variables analyzed here include extensive aerosol optical properties that depend on aerosol amount, absorption ($\sigma_{ap}$) and scattering ($\sigma_{sp}$) coefficients, as well as intensive aerosol optical properties, single scattering albedo (SSA) and scattering Ångström exponent (SAE), that are independent of the aerosol amount. Intensive aerosol properties presented in this analysis were calculated from extensive aerosol optical property measurements.

SAE describes the wavelength dependence of aerosol light scattering coefficient, and is inversely related to aerosol size such that large aerosols have small SAE values and vice versa (Delene and Ogren, 2002). SAE is calculated using Eq. (2):

$$SAE = -\frac{\log(\sigma_{s1}) - \log(\sigma_{s2})}{\log(\lambda_1) - \log(\lambda_2)} \qquad (2)$$

where $\sigma_{s1}$ is the light scattering coefficient at wavelength $\lambda_1$, and $\sigma_{s2}$ is the light scattering coefficient at wavelength

$\lambda_2$.

SSA is the ratio of scattering to extinction, as given in Eq. 3, and is indicative of aerosol darkness such that white aerosols (e.g., sea salt) have high SSA values and dark aerosols (e.g., black carbon) have low SSA values. SSA is calculated using Eq. (3):

$$SSA = \frac{\sigma_{sp}}{\sigma_{sp} + \sigma_{ap}} \qquad (3)$$

Aerosol asymmetry parameter, g, is a representation of the angular distribution of light scattering by an aerosol particle. The value of g can range from -1 to 1, depending on if light is entirely backscattered or forward scattered, respectively. Large particles have higher asymmetry parameters, indicating strong forward scattering. A value for g can be estimated using the backscatter fraction, b, which represents the fraction of backscattering to total scattering. Since the nephelometer measures backscattering and total scattering, b can be computed from nephelometer output.

Here, g is computed using Eq. (4), from Andrews et al. (2006) which was derived from an empirical fit to Fig. 3 in Wiscombe and Grams (1976):

$$g = -7.143889*b^3 + 7.464439*b^2 - 3.96356*b + 0.9893 \qquad (4)$$

All data were adjusted to common wavelengths (467, 525, 550, and 637 nm) for comparison among stations. For the nephelometers and absorption instruments with multiple wavelengths, Ångström exponents were used for the





wavelength adjustment. For single wavelength absorption instruments, a $1/\lambda$ relationship (Ångström exponent = 1) was assumed for wavelength adjustments. The scattering or absorption coefficient was then adjusted using Eq. (5):

$$\sigma_2 = \sigma_1 * (\frac{\lambda_1}{\lambda_2})^{AE} \qquad (5)$$

where $\sigma_1$ is the measured scattering or absorption coefficient at the instrument's native wavelength $\lambda_1$, $\sigma_2$ is the

scattering or absorption coefficient adjusted to the desired wavelength $\lambda_2$, and AE is the Ångström exponent (SAE for scattering coefficients, AAE for absorption coefficients). Wavelengths of measurements used are specified in all analyses presented here.

Uncertainties in PSAP and CLAP-measured absorption coefficient measurements come from instrumental noise, unit-to-unit variability, and instrument calibration, with a total measurement uncertainty of ~20-60% (Sherman et

al., 2015; Ogren et al., 2017). Uncertainties in Aethalometer absorption coefficient measurements depend on instrumental noise, instrument calibration, and flow controller performance. The total uncertainty of the measurements depends on monitoring station, attenuation, and Aethalometer wavelength channel (Backman et al., 2017). Uncertainties in MAAP-measured absorption coefficients stem from suitability of the selected asymmetry parameter to the sampled aerosol population, uncertainty in multiple scattering of the filter, and uncertainty in

diffuse fraction to yield a total uncertainty of 12% (Petzold and Schönlinner, 2004). Uncertainties in scattering coefficient measurements stem from instrumental noise, variability in nephelometer calibration, correction to STP, correction for angular non-idealities, and correction to RH<40% for when samples have higher humidity (if applicable) and give a total uncertainty of 8% (Sherman et al., 2015). More detailed information on measurement uncertainties in nephelometers, PSAP and CLAP data can be found in Sherman et al. (2015) and Ogren et al. (2017),

details on uncertainties in Aethalometer measurements can be found in Backman et al. (2017), and uncertainties in MAAP measurements can be found in Petzold & Schönlinner (2004).

## 4. Results and Discussion

### 4.1 Spatio-temporal variability of aerosol optical properties in the Arctic

The seasonality of aerosol light scattering ($\sigma_{sp}$) at the six monitoring stations reveals a diversity in magnitude and

seasonality of aerosol scattering across the Arctic. Figure 3 shows monthly median values of aerosol scattering coefficient, in Mm[-1], throughout the year at each station, as well as boxplots showing 5[th], 25[th], 50[th], 75[th] and 95[th] percentiles of hourly averaged scattering data for all months at each station. Aerosol scattering shows a strong seasonality at all sites in the study, though the seasonal cycle is not the same at each of the stations. Most sites (ALT, BRW, TIK, ZEP) show a scattering peak in the late winter and early spring, coincident with the Arctic Haze

phenomenon (Shaw, 1995; Quinn et al., 2007). These findings agree with many previous studies. At BRW, scattering data show a strong seasonality with values that are highest in the winter and spring during Arctic Haze season, and lowest in late summer (Bodhaine, 1983; Bodhaine; 1995; Delene and Ogren, 2002; Quinn et al., 2007). At ZEP, a study from several decades ago also finds higher scattering coefficients in the winter and lower scattering



coefficients in the summer (Heintzenberg, 1982), and a study by Pandolfi et al. (2017) is also consistent with the ZEP $\sigma_{sp}$ seasonal cycle presented here quite closely. The two other Arctic sites in this study exhibit distinctly different seasonal cycles. PAL measures maximum scattering coefficients in the summer, and minimum scattering values in the winter, opposite of what is observed at the first four stations. This finding agrees with previous

scattering climatologies at PAL from Aaltonen et al. (2006), Aalto et al. (2002), Hatakka et al. (2003), Lihavainen et al. (2015) and Pandolfi et al. (2017). In winter the scattering values at PAL are similar to values observed at ALT, BRW, TIK and ZEP, but in summer PAL measures notably higher scattering. PAL is located at the lowest latitude of all the sites in the study, and is the closest in proximity to the European continent. Although the site itself is located on top of a fell above the tree line, the station is surrounded by a forest, and thus affected by nearby biogenic

emissions during summer active vegetation season (Tunved et al., 2006; Lihavainen et al., 2009; Asmi et al., 2011). SUM is the highest in elevation of all the sites and measures free tropospheric air much of the year. This is reflected in the substantially lower scattering measurements made at SUM compared to the other stations. The seasonal cycle of scattering at SUM also differs from the other five Arctic sites considered here, in that it has a bimodal distribution of scattering, with a peak in early spring around April, and then another peak in late summer around August. There

is no signature of the Arctic haze phenomenon in the Summit aerosol optical property data, which is in agreement with previous radionuclide tracer studies performed at the site (Dibb, 2007). Annual statistics, including geometric mean, median, 25[th] percentile and 75[th] percentile, of aerosol light scattering coefficient are listed in Table 2 for each monitoring site.

The scattering coefficient boxplots for each station in Fig. 3 show that the spread of scattering data is generally

greatest during months when the scattering coefficient values are highest at each station. In other words, at ALT, BRW, TIK, and ZEP, the winter months have the largest range of scattering values (and the largest median scattering values), while the summer months have a smaller range of scattering values (and also the lowest median scattering values). This indicates larger day-to-day aerosol variability during the Arctic Haze season at these sites. PAL and SUM see larger spread of the scattering data during summer when scattering values are the highest.

Episodic long range transport of biomass burning aerosol (i.e., smoke), in addition to long range transport of anthropogenic aerosol from Europe and regional biogenic emissions, are likely contributing factors to the higher summer scattering values and spread of the data at these stations (Stohl et al., 2006a; Stohl et al., 2007; Hyvärinen et al, 2011). Other contributing factors likely include long range transport of anthropogenic aerosol from Europe as well as biogenic emissions (Hyvärinen et al., 2011; Tunved et al., 2006). In addition, at PAL, there is increased

contribution from continental air masses during the summer, which contribute to the higher scattering values (Aalto et al., 2002; Asmi et al., 2011).

Figure 4 shows monthly median values of aerosol light absorption coefficient ($\sigma_{ap}$) from corrected Aethalometer data at all six Arctic sites, as well as boxplots of absorption coefficients for all months. There is a robust annual cycle in aerosol light absorption at all of the Arctic stations. Most of the sites, including ALT, BRW, TIK and ZEP,

measure an absorption maximum in the late winter and early spring, coincident with scattering maxima and the Arctic haze season, and the lowest absorption values are measured in the summer months. This finding is in line



with previous publications that find climatology of black carbon concentrations or absorption coefficients with maxima in the spring and minima in the fall (Hopper et al., 1994 (ALT); Sharma et al., 2004 (ALT); Sharma et al., 2006 (ALT); Bodhaine, 1995 (BRW); Heintzenberg, 1982 (ZEP); Eleftheriadis et al., 2009 (ZEP)). As with scattering coefficients, these stations have greatest spread in absorption data during months where absorption

medians are highest. Of all the Arctic sites here, TIK has the highest absolute absorption coefficients during the winter, while PAL has the highest absorption coefficients during the summer. PAL and SUM again have slightly different absorption seasonality from the rest of the sites. PAL measures maximum aerosol light absorption in the winter, with much lower values in the summer, though the summer minimum was higher than at all other stations, likely due to the closer proximity and Europe and thus potential for long range transport. PAL notably has very large

variability in absorption during the months of December, January, and February, as seen in the boxplot of absorption at PAL in Fig. 4. SUM, the most remote and highest elevation site, shows a different cycle with its lowest absorption values in the winter and highest values in the summer, similar to the seasonality of scattering coefficients. Statistics, including geometric mean, median, 25$^{th}$ percentile and 75$^{th}$ percentile, of aerosol light absorption coefficient are listed in Table 2 for each monitoring site.

Single scattering albedo (SSA) values show seasonality at all of the Arctic sites. Figure 5 displays monthly median values of SSA, as well as boxplots of SSA for all months and all sites. ALT has relatively constant SSA values throughout most of the year, though SSA drops during July, coincident with large variability in SSA values as seen in the ALT boxplot. The SSA values at BRW are highest in the fall (September and October), and are otherwise fairly consistent the rest of the year, with the largest spread in SSA during months other than September and

October. The multi-year annual average of SSA at BRW was found to be 0.960 (see Table 2), which agrees with the SSA averages of 0.96 presented for BRW data from 1988-1993 in Bodhaine (1995) and 1997-2000 in Delene and Ogren (2002). PAL has higher SSA values in the summer and lower SSA values in the winter. This is explained by the seasonalities of absorption and scattering; lower SSA values occur in the winter when scattering is low and absorption is high, and higher SSA values occur in the summer when absorption is low and scattering is high. Aalto

et al. (2002) find that there is an increased contribution from continental air masses in the summer at PAL. Lihavainen et al. (2015) show that SSA in summer increases especially in continental air masses, although it is the highest throughout the year in marine air masses. The high SSA in summer is related to increasing biogenic contribution and decreasing contribution from anthropogenic sources, such as residential wood burning. SUM has similar SSA values throughout the year, except for when SSA drops to a median of 0.890 in September- quite a bit

lower than the annual median SSA of 0.954. Much of the increased summer operations are winding down at SUM around September, and the related increase in flights and transportation activities at this time could contribute to the lower SSA value during September. However, no instances in the data suggest contamination spikes that need removal; rather, we speculate that the increased anthropogenic activity at SUM at this time might contribute to a darker background aerosol. TIK has the most pronounced seasonal cycle in SSA, with median values of SSA around

0.860 in the winter, and higher SSA median values around 0.960 during the summer. TIK measures the darkest aerosol of all six Arctic stations during the winter. We speculate this could be due to an inversion layer trapping regional combustion aerosol produced from anthropogenic activities, energy production and transport, mainly in the




town of Tiksi and nearby villages. ZEP does not have a very distinguishable seasonality in SSA, though SSA values tend to be slightly lower during Arctic Haze season. The boxplots of SSA at ZEP indicate large variability in the SSA data at this station.

Scattering Ångström exponent (SAE) for the 450/700 nm wavelength pair is indicative of particle size, and has a
seasonal signature at only some of the Arctic stations. At ALT, the variability in SAE values is highest in the summer and fall months, suggesting that the site measures a variety of particle sizes during this time. However, the monthly median SAE does not show substantial change throughout the year. BRW does have seasonality in SAE, with lowest SAE values (larger particles) during the late summer and early fall, and higher SAE values in the spring (smaller particles). This same SAE seasonality at BRW was also observed in previous studies (Bodhaine, 1983;
Delene and Ogren, 2002), and one study offers an explanation to this seasonality with observations of an increase in sea salt when the sea ice melts in summer months (Quinn et al., 2002). PAL has a different seasonality with highest SAE values in the summer and lowest SAE values in the winter and early spring, which agrees with findings from Aaltonen et al. (2006) and Lihavainen et al. (2015). The statistics of SAE in Table 2 show an average SAE of 1.66 at PAL, which is close to the average of 1.7 +/- 0.7 that is reported in Lihavainen et al. (2015) and the median of 1.8
reported by Pandolfi et al. (2017). PAL and SUM statistical values of SAE are not directly comparable to the other Arctic sites due to their 2.5 μm size cut inlets, which limits measurements of large particles that would yield smaller SAE values. There is very little variability in SAE at SUM throughout the year, as the boxplot shows that medians of SAE in all months fall within the interquartile spread of SAE in all other months. However, it is notable that SUM generally has some of the highest SAE values of all six Arctic sites, meaning it is measuring some of the smallest
aerosol of these Arctic stations. These high SAE values are likely due to the remote high elevation location of SUM, which means larger particles fall out or are removed before reaching the monitoring station. Additionally, the long distance to the ocean from SUM means there is likely no sea salt measured, which can be a likely source for coarse aerosols in the Arctic. TIK has higher SAE values in March and October, with lower SAE values the rest of the year. Additionally, TIK sees the largest variability in SAE between the months of June and September. This large
variability could be attributed to Siberian wildfire events that occur sporadically during the summer, or to the secondary particle formation and growth by biogenic precursors that affect the site sporadically during the summer season (Asmi et al., 2016). Finally, ZEP measures smaller aerosols (larger SAE values) in the spring, and larger aerosols in the late summer, in accordance with the Arctic Haze phenomenon and in agreement with seasonal cycle of SAE at ZEP presented in Pandolfi et al. (2017). Mean SAE at ZEP (see Table 2) is 1.15, which is slightly higher
than the SAE median of just less than 1 presented in Pandolfi et al. (2017), which used ZEP data from years 2010-2014.

The variability of the asymmetry parameter, g, is similar for all sites except for SUM. Figure 7 shows that ALT, BRW, PAL, TIK and ZEP have highest values of g in the winter and lowest values in the summer. It is clear in these station subplots of Fig. 7 that the variability in g is largest during the summer months. This could be due to higher
noise in the nephelometer when scattering measurements are really low. In contrast, SUM shows the opposite seasonal cycle in asymmetry parameter, with g highest in the late summer and lowest in the late winter. PAL and




ZEP g seasonalities are in agreement with those presented in Pandolfi et al. (2017). To our knowledge, asymmetry parameter values for the other four sites have not been previously presented in the literature. However, Delene and Ogren did present the seasonality of backscatter fraction (b) for BRW, and because g is expected to vary inversely with b, and because they find that b is highest in the summer and lowest in the winter, their results are also

consistent with those reported here. Andrews et al. (2011) and Pandolfi et al. (2017) report a general tendency for g to increase as $\sigma_{sp}$ increases for mountain sites and European ACTRIS sites, respectively. The same tendency was found for the Arctic sites here (not shown). The lower g values at the 5 Arctic sites during the summer indicate the presence of smaller particles, probably due in part to wet scavenging of larger particles and/or new particle formation. Both processes tend to be more common in the summer (e.g., Freud et al., 2017) and are consistent with

the lower scattering coefficients observed in the summer. Higher g values throughout the rest of the year represent larger particles, perhaps due to long range transport. BRW, for example, has been impacted by Asian dust in the spring (e.g., Stone et al., 2007). However, SAE seasonality does not support this pattern at every site (the inconsistent relationship between SAE and g is also discussed in detail in Pandolfi et al. (2017)). This indicates that the specific shape of the aerosol size distribution at easch site will have a role in determining g and SAE at Arctic

sites as different aerosol parameters are sensitive to different parts of the size distribution (e.g., Collaud Coen et al., 2007).

**4.2 Systematic variability of aerosol optical properties in the Arctic**

The systematic variability of aerosol optical properties refers to how aerosol parameters co-vary with each other. Analysis of the systematic relationships between aerosol optical properties is useful because it can provide insight to

aerosol sources and atmospheric processes (Andrews et al., 2011; Toledano et al., 2007), and can also be a good metric for comparing consistency between aerosol models and measurements.

The systematic variability plots shown here were created by binning the hourly averages of aerosol light scattering coefficient values into 2 Mm$^{-1}$ bins between 0 and 20 Mm$^{-1}$ (this scattering range captures most of the station data (Fig. 3); a scattering coefficient of 20 Mm$^{-1}$ corresponds to the following percentiles at each station: 97.7 at ALT,

91.2 at BRW, 87.9 at PAL, 99.5 at SUM, 92.1 at TIK and 98.2 at ZEP) and then calculating and plotting median values of absorption coefficient, SAE, and SSA for each bin. This was repeated for 0.02 bins of SSA and plotting median values for SAE. As in Andrews et al. (2011), only bins that had a standard error (standard error is the standard deviation of the sample divided by the square root of the number of points in the sample) less than 2% of the typical value of that variable were included, with 2% of the typical values considered to be: ~0.02 for SSA,

~0.04 for SAE, and ~0.1 Mm$^{-1}$ for absorption coefficient. Bins with a larger standard error were omitted, since they may not be representative of actual aerosol systematic variability at the site.

Absorption coefficient varies with scattering coefficient almost linearly, such that absorption increases as scattering increases as shown in Fig. 8(a). One interpretation of this linear relationship between these scattering and absorption coefficients is that the scattering and absorbing aerosols are coming from the same sources, and are subject to

similar removal processes during transport to the site. This is consistent with systematic variability analysis from




Andrews et al. (2011) that looked at data from mountain sites. Delene and Ogren (2002) also show this systematic variability for BRW, over the same scattering range (0 - 20 Mm$^{-1}$) shown here, though Delene and Ogren (2002) find that absorption at BRW decreases at scattering values above ~20 Mm$^{-1}$. Higher scattering values were not investigated here.  Up to a scattering coefficient of about 8 Mm$^{-1}$, most of the stations (except for PAL), have a very

similar ratio of absorption to scattering, especially at lower absorption and scattering coefficients. This could be representative of a background Arctic aerosol being measured at all stations during relatively clean conditions. Where the ratios differ between stations at higher scattering and absorption values, a variety of local or long range transport sources could be influencing each station differently and changing this ratio. PAL looks different than other stations at the low loadings, where there is a higher ratio of absorption to scattering. Above a scattering

coefficient of 8 Mm$^{-1}$, ALT and TIK show a different systematic variability than the other stations, where ALT has a higher absorption to scattering ratio, and TIK has a much higher absorption to scattering ratio at high aerosol loadings. This suggests that at TIK high aerosol concentration events are strongly influenced by absorbing aerosols, which is consistent with the finding of Asmi et al. (2016).

Single scattering albedo varies with scattering such that the lowest scattering coefficient bins are accompanied by

relatively low SSA values, and SSA values plateau with higher scattering values- see Fig. 8(b). This finding follows the same pattern but with a much weaker dependence than what was found for mountain sites in Andrews et al. (2011), and shows a much weaker relationship than what was found for continental North American sites in Sherman et al. (2015). It should be noted that comparisons with systematic variability relationships for other site types are difficult since this Arctic analysis only looks at scattering from 0-20 Mm$^{-1}$, while the aforementioned

papers analyze a much greater range of scattering coefficients. The SSA vs. scattering relationship here suggests that whiter aerosols are preferentially scavenged such that darker aerosol remain at the lowest aerosol loadings (lowest scattering coefficients). Delene and Ogren (2002) find that SSA at BRW decreases slightly between scattering coefficient bins between 0-10 Mm$^{-1}$, but SSA increases after that as scattering increases. TIK looks different from the other Arctic sites since SSA increases with scattering only up until a scattering coefficient of ~5 Mm$^{-1}$, after

which SSA decreases. This means higher aerosol loadings at TIK have darker aerosol, which could be representative of fresh smoke emissions affecting the site at high aerosol loadings, in accordance with the systematic variability of absorption with scattering.

Scattering Ångström exponent varies with scattering in diverse ways at the six Arctic stations, as indicated in Fig. 8(c). At sites like ALT, SUM and TIK, SAE does not vary much with changes in scattering. BRW generally shows

decreases in SAE (or increases in particle size), as scattering increases. Delene and Ogren (2002) show that the aerosol particles at BRW tend to be largest (lowest SAE) and whitest (highest SSA) during the summer (lowest scattering values), which they attribute to the contribution of marine aerosol when the sea ice melts. Chemical analysis has supported this conclusion (Quinn et al., 2002), though the systematic variability plots shown here do not provide the means to analyze this seasonality. PAL and ZEP show distinctly different systematic variability from

BRW, in that SAE increases (decreasing particle size) as scattering increases.



Figure 8(d) shows that SAE also varies with SSA. At ALT, BRW, and ZEP, SAE decreases as SSA increases. This indicates that the more scattering particles are typically larger at these sites (e.g., sea salt), and more absorbing particles are typically smaller (e.g., black carbon). There are not enough data that meet the threshold to detect systematic variability in these properties at TIK. PAL and SUM do not show substantial systematic variability in these optical parameters, likely due to their 2.5 μm size cut inlet (PAL and SUM) and/or remote high elevation location (SUM) that limits the measurement of larger particles and thus yields consistently high SAE values.

**4.3 Back-trajectory analysis**

Back-trajectory analyses are widely used to investigate the effect of air mass pathway on atmospheric constituents measured at a particular place (Fleming et al., 2012). The trajectory method involves calculating air parcel movement from the monitoring site back in time to yield the back-trajectory of the parcel (Draxler and Hess, 1998). Here, individual 7-day back-trajectories computed for each of the six Arctic sites are overlaid and colored by frequency of back-trajectory occurrence in each grid box to create a density plot of air mass history for each station.

In this work, the air-mass back-trajectory analysis was conducted using the NOAA Hybrid Single-Particle Lagrangian Integrated Trajectory model (HYSPLIT) version 4.9 (Draxler and Hess, 1998; Stein et al., 2015). The HYSPLIT model was run for 7-day back trajectories, using an ensemble method. The ensemble method offsets the meteorological grid by one grid point in the horizontal and 1% of the surface pressure in the vertical, which produces 27 back-trajectories for possible offsets in the horizontal and vertical, thus accounting for uncertainties in the gridded meteorological data. The meteorological data used for the trajectories was the NCEP/GDAS dataset with a 1° horizontal resolution and 23 pressure levels (Kanamitsu, 1989).

Figure 9 shows density plots of each 7-day back-trajectory path computed at each station over the period of interest (2012-2014), colored by frequency at which the air mass passed through the given grid cell. Regions colored in red represent regions through which air masses most frequently traveled en route to the monitoring station, and regions colored in blue represent areas through which an air mass passed least frequently en route to the monitoring station. All trajectory altitudes are included in plots in Fig. 9.

For all measurement sites, air masses arriving at the site obviously pass most frequently through regions closest to the stations. The differences between summer and winter back trajectories at each site are subtle, and do not reflect the large seasonality observed in aerosol optical property measurements throughout the Arctic. This is consistent with similar back trajectory frequency analyses at ALT (Sharma et al., 2006; Huang et al., 2010). This could be because the wide range of synoptic-scale weather patterns averaged into three years of back trajectory data obscure seasonality in large-scale air mass paths. One feature that is evident from Fig. 9 is that SUM does not seem to have the same air mass origin as the other sites. Even the closest station, ALT, does not overlap much with calculated source areas for SUM. This feature is even more clear when only trajectory altitudes below 500 m agl are considered. This supports the earlier argument that, due to the altitude and location of SUM on top of the Greenland ice shelf, the aerosol arriving at the stations is very different compared to the other sites that are almost exclusively



coastal. The strong seasonality observed in the aerosol optical properties at each of the Arctic sites is likely not due to large changes in air mass back trajectories from season to season. If the seasonality of the aerosol parameters is not described by differences in air mass origin, then we speculate that the aerosol sources (both natural and anthropogenic) differ in type and magnitude from season to season and may explain the temporal variability of

aerosols in the Arctic. This notion is supported by previous studies (Eleftheriadis et al., 2009; Asmi et al., 2016; Wang et al., 2014; Sharma et al., 2013). Alternatively, it is possible that much longer back trajectories would elucidate additional information on seasonal differences in air mass origin for long distance aerosol transport to the Arctic (Qi et al., 2017). For example, work by Hirdman et al. (2010) uses 20-day back trajectories from FLEXPART and suggests stronger seasonal differences in aerosol transport pathways than was found here. Using much longer

back trajectory calculations in this study would, however, also be associated with much greater uncertainties in the spatial domain, which is why the trajectory calculations were restricted to 7 days.

For further exploration of why aerosol sources (rather than transport) might differ in type and magnitude from season to season, Figure 10 affords insight into how the land type over which an air mass travels might affect the aerosols within it. Figure 10 shows the percent of air mass residence time spent above different land types before

arriving at each monitoring station for each month of the year. The data used for sea ice extent came from the National Snow & Ice Data Center's Sea Ice Index data set (Fetterer et al., 2015). The green bars represent land (with no distinction between snow-covered and bare land areas), light blue bars represent sea ice, and dark blue bars represent open water. There is a clear seasonality in land type over which air masses travel before arriving at each measurement site. At all sites except SUM, air masses travel more over open water during the summer when sea ice

has melted. This provides a source for sea salt and other marine aerosol during the summer that is much less likely at other times in the year. The result that the same source region overlaps with open ocean in summer and sea ice in winter, and thus yields different aerosol, is support by similar findings from Shaw et al. (2010). TIK, PAL and SUM are similar in that most of the air mass residence time is spent above land at all times of the year, but especially so in winter. ALT, ZEP and BRW are similar in that the air masses arriving at these stations spent more time, compared to

the other sites, over sea ice and much less time over land. This could explain why ALT, ZEP and BRW have very similar seasonality of aerosol light scattering and absorption coefficients, while TIK, PAL and SUM have different seasonality that may be indicative of varying land-based aerosol sources. More work is needed, using chemical analyses or footprint analyses, to better understand how air mass transport contributes to the different aerosol seasonality at each of the six Arctic sites.

**5. Conclusions**

Seasonal cycles of aerosol optical properties from six Arctic monitoring stations have been presented here. Aerosol optical properties were derived from common absorption and scattering instruments (Aethalometers and nephelometers, respectively) at the stations, were evaluated and corrected under common quality control procedures, and were presented at standard temperature and pressure and low relative humidity to ensure high quality and

comparability of data across stations.





The extensive aerosol optical properties, dependent on amount of aerosol, showed strong seasonality at all of the Arctic sites analyzed here. The magnitude and variability of aerosol light scattering coefficient varies substantially between stations, with SUM measuring the lowest annual mean scattering coefficients (1.74 Mm$^{-1}$) and TIK measuring the highest annual mean scattering coefficients (12.47 Mm$^{-1}$). ALT, BRW, TIK and ZEP have maximum

scattering values in the spring, and lowest in the summer, while PAL and SUM have lowest scattering values in the winter and highest in the summer. The magnitude and variability of aerosol light absorption coefficient is slightly less variable between stations compared to scattering. The lowest annual mean absorption coefficient is measured at SUM (0.12 Mm$^{-1}$), while the highest annual mean absorption coefficient is measured at PAL (0.48 Mm$^{-1}$). Stations ALT, BRW, PAL, TIK and ZEP all have a seasonal cycle that reflects high absorption in the winter and spring, and

low absorption in the summer, though the exact timing of the absorption maxima and minima differs among stations. SUM absorption is unique from the other sites in that the highest absorption values are in summer, and lowest absorption values are in winter. The distinctiveness of the SUM seasonality is likely due to its remote and high elevation location.

The intensive aerosol optical properties, which are independent of aerosol amount, also show strong seasonality at

all six Arctic stations. Furthermore, quite high SSA values at all stations are evident in our data. The range of annual mean single scattering albedo values at the sites is from 0.909 at PAL to 0.960 at BRW. The annual mean scattering Ångström exponent values range from 1.04 at BRW to 1.80 at SUM. The annual mean aerosol asymmetry parameter values range from 0.57 at ALT to 0.75 at SUM.The seasonalities of these variables suggest that aerosol source and removal mechanisms are likely different from month-to-month at a given site, and from site-to-site throughout the

Arctic.

Systematic variabilities of the aerosol optical parameters measured in the Arctic provide insight into atmospheric processes near the monitoring stations. Generally, absorption coefficients increase as scattering coefficients increase at all of the sites. However, the ratio of absorption to scattering is different across sites and aerosol loadings, with TIK and ALT showing higher absorption to scattering ratios at high aerosol loadings, and PAL showing higher

absorption to scattering ratios at low aerosol loadings compared to the other stations. Single scattering albedo is low at low loadings for all of the six Arctic sites, and SSA increases with increasing scattering for most sites. TIK is an exception to this observation, since darker aerosol (low SSA) is measured at higher scattering coefficients, which suggests absorbing aerosol (e.g., black carbon) may be associated with high aerosol loading events (e.g., anthropogenic emissions, Siberian wildfires). Our findings of generally higher aerosol absorption and lower SSA for

both TIK and PAL during winter could suggest a closer proximity to anthropogenic activities, which is supported by their geographic locations since they are both continental Eurasian locations- closer to forest fires, long-range transport, and regional emissions.

Back trajectory analysis showed little evidence of seasonality in air mass origin between winter and summer months. The analysis further strengthens the observation that SUM is different from the other stations because other stations

seem to receive little air from the same areas that SUM does. Data on sea ice combined with air mass movement




indicated that TIK and PAL receive the most continental airmasses whereas BRW, PAL, and ZEP are the stations with the potential to be most influenced by marine aerosol.

A persistent and important theme in the findings of this paper is that aerosol optical properties vary widely with season at any individual site, and they vary widely from station to station throughout the Arctic. This result is

important, since it means that the Arctic cannot be treated as a uniform region, spatially or temporally, in climate models or in remote sensing retrieval algorithms. Rather, the wide spatio-temporal variability of aerosol in the Arctic needs to be considered in order to properly represent the climate of this sensitive region.

**Data availability**

Data used in this article are archived and accessible from the EBAS database operated at the Norwegian Institute for Air Research (NILU) (http://ebas.nilu.no).

**Competing interests**

The authors declare that they have no conflict of interest.

**Acknowledgements**

Thank you to all of the station technicians at these Arctic monitoring sites who work in difficult Arctic conditions to help acquire the data presented here. The authors would like to acknowledge the International Arctic System for Observing the Atmosphere (IASOA) aerosol working group for coordination of the project and contribution of

expertise to this analysis. Data management is provided by the WMO Global Atmosphere Watch World Data Centre for Aerosol. This project has received funding from the European Union's Horizon 2020 research and innovation programme under grant agreement No 654109 (ACTRIS). The Finnish Meteorological Institute acknowledges the Academy of Finland project Greenhouse gas, aerosol and albedo variations in the changing Arctic (project number 269095), the Novel Assessment of Black Carbon in the Eurasian Arctic: From Historical Concentrations and

Sources to Future Climate Impacts (NABCEA) (project number 296302), the Academy of Finland Centre of Excellence program (project number 307331), and EU H2020 Project INTAROS (Project ID: 727890) for financial support. Funding from the NOAA Climate Program Office provided partial support for data analysis and measurements at Barrow and Summit. The authors would like to thank the staff of Canadian Forces Service for maintenance of Alert station. The light scattering measurements at Alert were initiated by Richard Leaitch.



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



Table 1. Monitoring station names, locations, scattering and absorption instruments, size cuts and humidity of samples. Bolded instruments indicate those from which data is used in this analysis.

| Station Code & Location | Latitude / Longitude / Elevation | Scattering Instrument(s) | Co-located Absorption Photometer Instrument(s) | Aethalometer Model(s) | Size Cut (μm) | RH of Sample |
|---|---|---|---|---|---|---|
| **ALT** Alert, Canada | 82.49915°N 62.34153°W 210 masl | *2004-present:* **Nephelometer 3563^** | *2004-2010:* PSAP-1W§ *2007-present:* **PSAP-3W†** | *1989-2009: Aethalometer AE6‾* **2008-present: Aethalometer AE31■** | 1 **10** NA | < 40% |
| **BRW** Barrow, Alaska | 71.32301°N 156.6115°W 11 masl | *1976-1997:* Nephelometer 1559B* *1997-present:* **Nephelometer 3563^** | *1997-2006:* PSAP-1W§ *2006-present:* PSAP-3W† *2011-present:* **CLAP** | *1988-2002:* Aethalometer AE8△ **2010-present: Aethalometer AE31■** *2014-present:* Aethalometer AE33▽ | 1 **10** **NA** | < 40% |
| **PAL** Pallas, Finland | 67.97361°N 24.11583°E 560 masl | *2000-present:* **Nephelometer 3563^** | *2007-present:* **MAAP°** | **2005-present: Aethalometer AE31■** | **2.5** NA | < 40% |
| **SUM** Summit, Greenland | 72.58000°N 38.48000°W 3238 masl | *2011-present:* **Nephelometer 3563^** | *2011-present:* **CLAPχ** | **2003-present: Aethalometer AE16©** *2014-present:* Aethalometer AE33▽ | **2.5** | < 40% |
| **TIK** Tiksi, Russia | 71.58617°N 128.91882°E 8 masl | *2013-present:* **Nephelometer 3563^** | *2013-present:* **MAAP°** | **2009-present: Aethalometer AE31■** | **10** | < 30% |
| **ZEP** Zeppelin Mountain, Ny-Ålesund, Norway | 78.90669°N 11.88934°E 475 masl | *2010-present:* **Nephelometer 3563^** | *2002-present:* **PSAP-1W♦** | **2005-present: Aethalometer AE31■** | NA | < 20% |

^TSI Nephelometer 3563
*MRI Nephelometer 1559B
§Radiance Research 1-wavelength Particle Soot Absorption Photometer (PSAP-1W)
♦Custom built 1-wavelength Particle Soot Absorption Photometer (PSAP-1W)
†Radiance Research 3-wavelength Particle Soot Absorption Photometer (PSAP-3W)
χNOAA Continuous Light Absorption Photometer (CLAP)
° Thermo Fisher Scientific Multi-angle Absorption Photometer (MAAP) Model 5012
‾Magee Aethalometer AE6
△Magee Aethalometer AE8
■Magee Aethalometer AE31
▽Magee Aethalometer AE33
©Magee Aethalometer AE16



Table 2. Statistics of aerosol optical properties at six Arctic monitoring sites, including geometric means, medians and interquartile spread of absorption coefficient ($\sigma_{ap}$) at 550nm, scattering coefficient ($\sigma_{sp}$) at 550nm, single scattering albedo (SSA) at 550nm, and scattering Ångström exponent (SAE) at 450/700nm. Percentile statistics are based on hourly averages.

| Variable | Statistic | ALT | BRW | PAL | SUM | TIK | ZEP |
|---|---|---|---|---|---|---|---|
| $\sigma_{ap}$ (Mm$^{-1}$) | Geometric Mean | 0.30 | 0.30 | 0.48 | 0.12 | 0.74 | 0.18 |
| | 25th percentile | 0.07 | 0.08 | 0.12 | 0.02 | 0.12 | 0.04 |
| | 50th percentile | 0.20 | 0.20 | 0.24 | 0.05 | 0.43 | 0.09 |
| | 75th percentile | 0.41 | 0.39 | 0.49 | 0.11 | 0.98 | 0.23 |
| $\sigma_{sp}$ (Mm$^{-1}$) | Geometric Mean | 5.61 | 8.89 | 9.18 | 1.74 | 12.47 | 4.35 |
| | 25th percentile | 1.18 | 3.03 | 1.95 | 0.26 | 2.19 | 1.19 |
| | 50th percentile | 4.11 | 6.93 | 4.74 | 0.80 | 6.06 | 2.82 |
| | 75th percentile | 8.31 | 12.05 | 10.97 | 1.93 | 10.88 | 5.53 |
| SSA (dimensionless) | Geometric Mean | 0.929 | 0.960 | 0.909 | 0.913 | 0.934 | 0.945 |
| | 25th percentile | 0.927 | 0.948 | 0.907 | 0.917 | 0.908 | 0.940 |
| | 50th percentile | 0.949 | 0.969 | 0.956 | 0.954 | 0.950 | 0.963 |
| | 75th percentile | 0.965 | 0.984 | 0.976 | 0.973 | 0.972 | 0.980 |
| SAE (dimensionless) | Geometric Mean | 1.18 | 1.04 | 1.66 | 1.80 | 1.56 | 1.15 |
| | 25th percentile | 0.85 | 0.58 | 1.22 | 1.41 | 1.30 | 0.64 |
| | 50th percentile | 1.21 | 1.02 | 1.81 | 1.93 | 1.70 | 1.24 |
| | 75th percentile | 1.50 | 1.48 | 2.17 | 2.35 | 2.03 | 1.69 |
| g (dimensionless) | Geometric Mean | 0.57 | 0.61 | 0.64 | 0.75 | 0.58 | 0.59 |
| | 25th percentile | 0.54 | 0.58 | 0.53 | 0.41 | 0.53 | 0.52 |
| | 50th percentile | 0.60 | 0.63 | 0.60 | 0.61 | 0.59 | 0.57 |
| | 75th percentile | 0.64 | 0.65 | 0.66 | 0.78 | 0.63 | 0.62 |





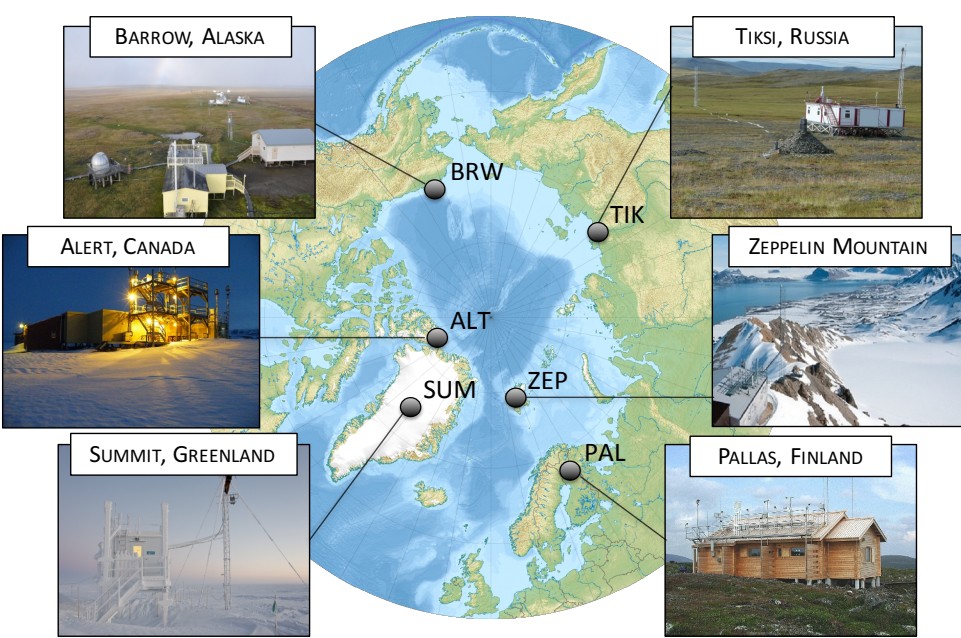

Figure 1. Map of Arctic monitoring stations with pictures of each site





Figure 2. Comparison of monthly median, averaged from hourly data and corrected Aethalometer absorption (light blue) and absorption measured by co-located absorption instrument (orange). (a) ALT absorption at 467nm, (b) BRW absorption at 467nm, (c) PAL absorption at 637nm, (d) SUM absorption at 880nm, (e) TIK absorption at 637nm, (f) ZEP absorption at 525nm. All data are at wavelength of co-located absorption instrument (PSAP, CLAP or MAAP), except for SUM where data are at wavelength of the 1-wavelength AE16 Aethalometer (880nm). Note that y-axes are different on each plot.







Figure 3. Seasonality of aerosol light scattering coefficient ($\sigma_{sp}$) at 550nm at all sites. Large plot shows monthly medians of scattering in Mm$^{-1}$ at each station, subplots below show boxplots of hourly average scattering at individual sites with horizontal line at the median, edges of the box at 25$^{th}$ and 75$^{th}$ percentiles, and whiskers at 5$^{th}$ and 95$^{th}$ percentiles. Note that y-axes are different on each plot.





Figure 4. Seasonality of aerosol light absorption coefficient ($\sigma_{ap}$) at 550nm at all sites. Large plot shows monthly medians of absorption in Mm$^{-1}$ at each station, subplots below show boxplots of hourly average absorption at individual sites with horizontal line at the median, edges of the box at 25$^{th}$ and 75$^{th}$ percentiles, and whiskers at 5$^{th}$ and 95$^{th}$ percentiles. Note that y-axes are different on each plot.





Figure 5. Seasonality of single scattering albedo (SSA) at all sites. Large plot shows monthly medians of hourly average SSA at 550nm at each station, subplots below show boxplots of SSA at individual sites with horizontal line at the median, edges of the box at 25[th] and 75[th] percentiles, and whiskers at 5[th] and 95[th] percentiles. Note that y-axes are different on each plot.



Figure 6. Seasonality of scattering Ångström exponent (SAE) at all sites. Large plot shows monthly medians of hourly average

5 SAE at the 450nm/700nm wavelength pair at each station, subplots below show boxplots of SAE at individual sites with

horizontal line at the median, edges of the box at 25[th] and 75[th] percentiles, and whiskers at 5[th] and 95[th] percentiles. Note that y-

axes are different on each plot.



Figure 7. Seasonality of aerosol asymmetry parameter (g) at all sites. Large plot shows monthly medians of hourly average g at 550nm at each station, subplots below show boxplots of g at individual sites with horizontal line at the median, edges of the box at 25th and 75th percentiles, and whiskers at 5th and 95th percentiles. Note that y-axes are difference on each plot.



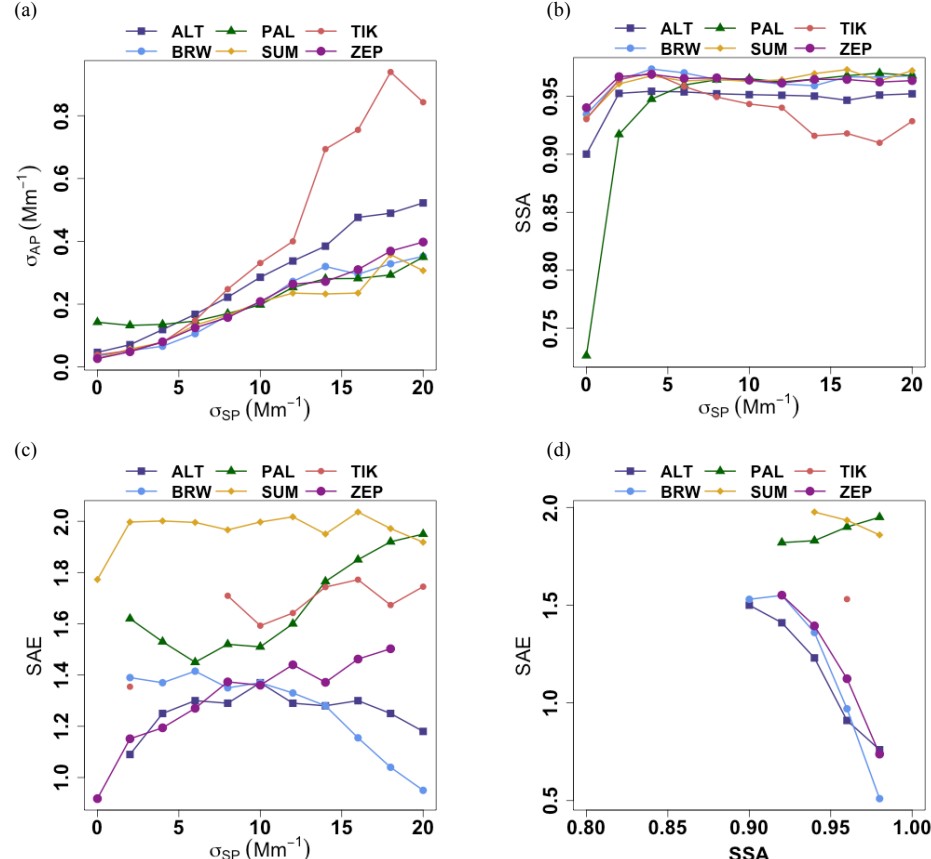

Figure 8. Systematic variability of median aerosol optical properties: (a) Absorption varying with scattering coefficient, (b) Single scattering albedo varying with scattering coefficient, (c) scattering Ångström exponent varying with scattering coefficient, and (d) scattering Ångström exponent varying with single scattering albedo.





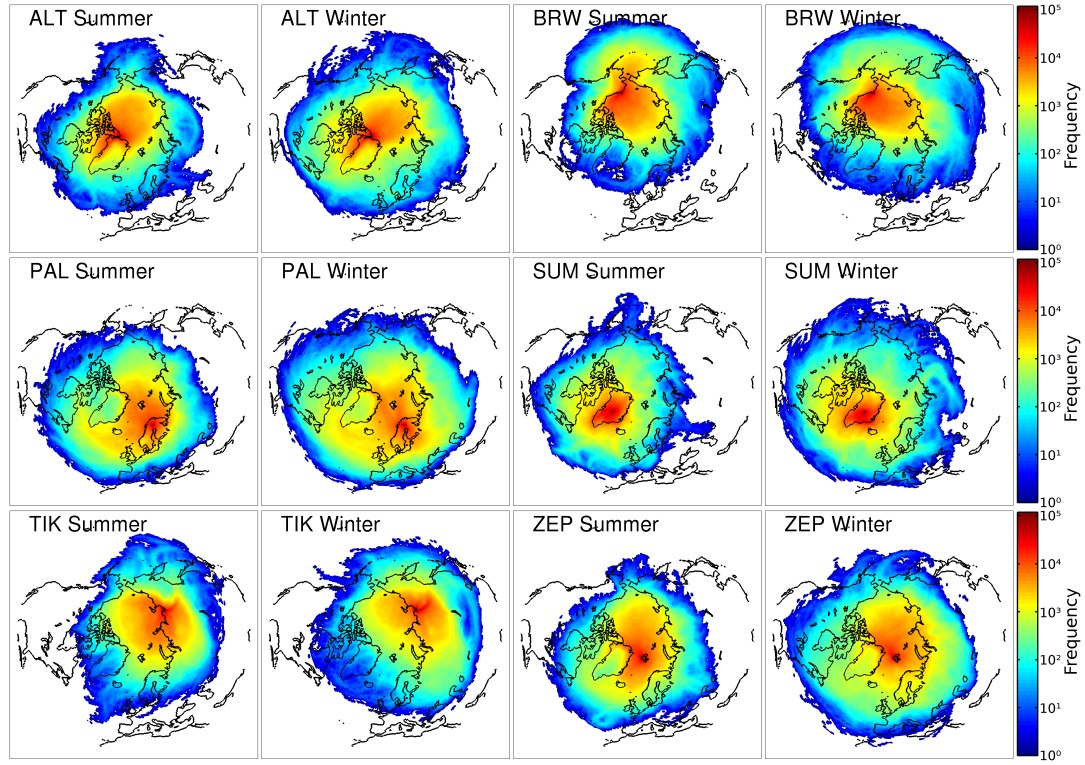

Figure 9. 7-day Back trajectories at each of the six Arctic stations, separated by summer (May-October) and winter (November-April) months. Colors represent frequency (units of hrs per 2 years) at which an air parcel travels over that region before arriving at the station; in other words, residence time of air in that location. These plots show data from all trajectory altitudes.





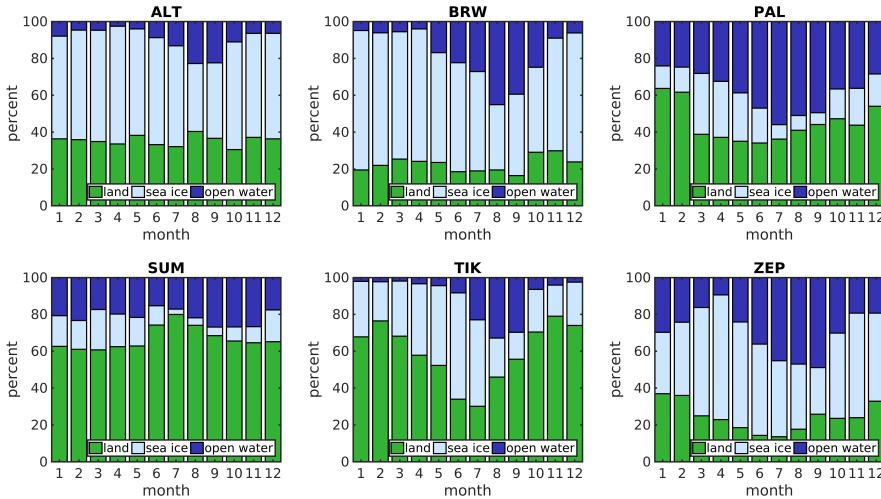

Figure 10. Percent of air mass residence time during the past 7 days spent above different land types before arriving at monitoring station for each month of the year. Green represents land (with no distinction between snow-covered and bare land areas), light blue represents sea ice, and dark blue represents open water.