# Peer review of "Seasonality of aerosol optical properties in the Arctic"

_Atmospheric Chemistry and Physics, 2017_

## Referee Comment (RC1) · Anonymous Referee #1 · 21 Feb 2018

This paper compares the seasonality of bulk aerosol optical properties (either sub-2.5 um or sub-10 um) at six sites in the Arctic for the years 2012 to 2014. The presentation of monthly median values for each site reveals differences in seasonality and boxplots of hourly averaged data reveal monthly-to-month variability. A main conclusion is that optical properties vary regionally across the Arctic primarily due to variability in source regions. The paper is straightforward and well-written. I recommend publication with only the minor corrections listed below.

Table 1. The information in the size cut column is confusing. Should it be aligned with the different instruments? What does NA mean?

Figure captions. I realize that the size range of the measurements is given in the text but it would be useful to provide it in the figure captions, too, since many readers may

only look at the figures.

Page 12, lines 12 - 14: What are the April and August peaks in light scattering at Summit due to?

Page 12, line 6: It is not clear from Figure 4 that PAL "has the highest absorption coefficients during the summer."

Page 13, lines 28 - 32: Could lower SSA at SUM during September be due to forest fires?

Figure 9. It would be helpful to have location markers for the stations on the maps.

---

## Referee Comment (RC2) · Anonymous Referee #2 · 27 Feb 2018

Review of the manuscript "Seasonality of aerosol optical properties in the Arctic" by Lauren Schmeisser et al.

This manuscript presents the seasonality of aerosol particles optical properties at six Arctic observatories. Aerosol scattering and absorption, scattering Angstrom exponent (SAE), single scattering albedo (SSA) and asymmetry parameter (g) are presented and discussed.

In the abstract and in the introduction the authors comment about the importance of studying the in-situ surface aerosol optical properties in the Arctic given the sensitivity of the Arctic climate to short-lived climate forcers.
In general, more speculations about the reasons explaining the observed differences among the stations are needed to improve the scientific significance of the presented work.
In most cases the manuscript presents a list of extensive and intensive values/properties at each station but the reasons behind the observed aerosol optical properties is sometimes missing.
The manuscript should be more focused on the Arctic haze phenomenon. For this, a reorganization of the manuscript is needed. Some suggestions are given below.

1) The six stations included in this work have two types of filter based absorption instruments: the "reference" instrument (CLAP, PSAP, or MAAP) and the Aethalometer model AE31. The AE31 attenuation data are corrected with the Arctic specific correction factor from Backman et al. (2017). The same $C_f$ ( = 3.20) is used to correct the AE31 data from the six observatories. Absorption data collected with the AE31 at 550 nm are presented in this manuscript. The absorption at 550 nm is calculated using the absorption Angstrom exponent (AAE) calculated from the 7-$\lambda$ aethalometer measurements. The authors show that the comparison between absorption from the "reference" instrument and the AE31 is "imperfect and variable among stations".

Why not present the absorption measurements from the "reference" instruments rescaled to 550 nm using the AAE from the AE31 instruments? If CLAP, PSAP or MAAP are considered as "reference" instruments, then data from these instruments should be presented in the manuscript. Moreover, the same $C_f$ is applied to the seven absorption measurements from the AE31 instruments thus meaning that the AAE from uncorrected AE31 data and the AAE from corrected data should be approximately the same.

2) Alternatively, if the authors think that the AE31 data are sufficiently robust to be presented in the manuscript (note that the supplemental material is not provided by the authors, so it is difficult to evaluate the goodness of the corrected AE31 data (and consequently SSA values)), then they should take more advantage of the multi-wavelength absorption measurements from AE31 instruments.

Is there any specific/interesting feature in the AAE seasonality at the six stations?
Why not study the spectral dependence of the single scattering albedo (SSA)? For example, presenting the SSA not only in the green, but also in the UV and near IR?
The variability of these two quantities (AAE at least) should be discussed in the manuscript.

3) In the manuscript the Arctic Haze (AH) phenomenon is discussed together with the scattering and absorption measurements. ALT, BRW, TIK, and ZEP stations present an increase in both scattering and absorption in late winter/spring related to the AH phenomenon.
However, there is no mention to the AH phenomenon in the sections presenting the intensive aerosol optical properties.

   a) A table presenting the mean SAE, SSA, $g$ (and possibly AAE) during AH period versus non-AH period should be presented and discussed. The spatial differences (from one site to another) in the intensive optical properties during AH period should be also discussed. For example, the seasonality of scattering and absorption at ALT, BRW, TIK and ZEP is very similar (and ascribed by the authors to the AH phenomenon) whereas the intensive properties are very different. For example, the SSA at ALT during AH is much higher (and different in term of seasonality) from the SSA observed at TIK during the AH period. The authors should comment/discuss the possible reasons explaining why the intensive properties change from one site to another during AH period.

   b) It is interesting the fact that the effect of AH on intensive properties is not observed at PAL and SUM which are located at higher altitude compared to the other stations. Is there any relationship between altitude of the station and AH phenomenon?

4) The authors say that "..surface Arctic aerosol optical properties in particular can help define and constrain inter-annual, seasonal and diurnal variability" (Pag. 2, Line 22-23). Why not present the diurnal cycles of both extensive and intensive aerosol particle optical properties? This can be done comparing AH period versus non-AH period.

5) Improve the abstract. In the present form the abstract present a list of lowest/highest values of extensive and intensive properties at the six observatories, but the reasons/speculations behind the variability of the reported values is missing.

6) Pag. 8, Line 31. Figure 2 shows the time series of monthly median corrected AE31 data. Why not present the daily median? Note also that the supplemental material was not uploaded. Consequently, it is very difficult to evaluate the goodness of the comparisons using just monthly medians.

7) Pag. 9, Line 33: The authors should explain where the data came from. For example, was it downloaded from EBAS. Or was it provided by data providers?

8) Pag. 10, Line 8: Add that also $g$ was one of the variables considered in the manuscript.

9) Pag. 10. How were the intensive properties calculated? Using all the scattering and absorption data or using only data above a given threshold (i.e. >1 Mm$^{-1}$)?. Calculating

the intensive properties using scattering or absorption data higher than a given threshold is important in order to remove undesired noise in the calculations.

For example: In Figure 2 the SSA at ALT and SUM in July and September, respectively, presents the lowest values when also scattering and absorption are low. The same is observed for the scattering Angstrom exponent at SUM in winter or the asymmetry parameter at SUM in January (for example).
How do these figures (Figures 5, 6, 7) change if a threshold is applied before calculating the intensive properties? In the case of SSA at SUM in September the authors speculate that the low SSA is related to an increase in flights and transportation activity. However, for other stations/seasons no explanations are given to justify why the $5^{th}$ and $95^{th}$ percentiles are too low or high. It is important to demonstrate that these high deviations of intensive properties at some stations are not due to noise.

10) Pag. 10, Equation 2: Why not present the differences between the SAE calculated between 450 and 550 nm and the SAE calculated between 550 and 700 nm? Is there any interesting difference between the two SAE during the AH phenomenon versus periods without AH phenomenon?

11) Pag. 10. The AAE from AE31 was used to calculate the absorption at the same wavelength of the "reference" instrument. How was the AAE calculated? Were used all the wavelengths or only those close to the reference wavelength?

Moreover, (end of Pag. 10 – beginning of Pag. 11), the authors say that the SAE was also used for the wavelength adjustment of nephelometer data. However, the TSI nephelometer works at 550 nm which is the wavelength used to present the results. So, no adjustment of nephelometer data is in principle needed. Please, clarify.

12) Pag. 13, Line 6. PAL -> SUM

13) Pag. 13, Lines 16-18: Explain why at ALT the SSA values drop during July (any physical explanation or noise?)

14) Pag. 13, Line 18: Explain why the SSA values at BRW are the highest during September-October.

15) Pag. 13, Lines 22-24 ("This is explained by ............   is low and scattering is high"). Remove the sentence. This is obvious.

16) Pag. 13, Line 25 and Lines 27-28: The high scattering at PAL in summer is probably due to the enhanced formation of BSOA. This is probably consistent with the fact that absorption does not show the same increase in summer. Consequently, the SSA is the highest in summer (with quite low standard deviation of the data) and reflects the

presence of very "white" particles. However, the authors say (Line 25) that there is an increased contribution from continental air masses in the summer at PAL. So, what is driving the evolution of the extensive and intensive properties at PAL in summer? The arriving of continental air masses (probably containing less "white" particles) or the BSOA formation (Lines 27-28)?

17) Pag. 14, Lines 24: Also here it is important to demonstrate that the large variability in SAE in July-September at TIK (when scattering and absorption are very low) is not due to noise. It is important to know if any threshold has been applied before calculating the intensive properties.

18) Pag. 14, Line 32: It seems that $g$ also varies quite a lot from one station to another, whereas the authors say that "the asymmetry parameter, $g$, is similar for all sites except for SUM". Please, clarify/expand.

19) Pag. 15, Line 7 and Figure 8: Why not show the $g$ too?

20) Table 2: SUM station registers the highest SAE (small particles) and also the highest $g$ (large particles). Any explanation for this?

21) Section 4.3: Figure 10 is nice. It seems that there is a relationship between the time spent above open water and sea ice and Figures 3 and 4. For example, at TIK the scattering is the lowest when air masses spent more time over sea ice and open water (June to September). At ZEP the reduction of scattering between June and October reflects the relative increase of time spent over sea ice and open water (and less time spent over land). Can the authors say something more about Figure 10? Is it possible to relate the time spent over land with the Arctic haze phenomenon? The paragraph at Pag. 18, Lines 16-29 should be expanded.

Figure 9 seems less useful. The highest frequency is always observed for regions close to the stations. Why not use, i.e., the potential source contribution function or the concentration weighted trajectory? (Both are available for example in the OPENAIR r package). These plots could be colored by levels of scattering and absorption to get a clearer idea about source regions. The differentiation in terms of air masses between AH periods versus non-AH periods should be introduced and discussed.

---

## Author Comment (AC1) · 10 May 2018

**Response to Anonymous Referee #1**

Thank you, anonymous referee #1, for your useful comments. The manuscript is improved thanks to your input.

Our response is structured as follows: original comments from reviewer #1 are bolded, our responses are in italics, and the revised portions of the manuscript follow in quotation marks with specific changes/additions in red.

**Comments**

**This paper compares the seasonality of bulk aerosol optical properties (either sub-2.5 um or sub-10 um) at six sites in the Arctic for the years 2012 to 2014. The presentation of monthly median values for each site reveals differences in seasonality and boxplots of hourly averaged data reveal monthly-to-month variability. A main conclusion is that optical properties vary regionally across the Arctic primarily due to variability in source regions. The paper is straightforward and well-written. I recommend publication with only the minor corrections listed below.**

**Table 1. The information in the size cut column is confusing. Should it be aligned with the different instruments? What does NA mean?**

*We agree that this column in Table 1 is confusing, thus we changed the format within the table so that the size cut is listed in brackets directly after the instrument used in this analysis. In addition, the text describes this size cut information in further detail.*

*Detailed information on the size cuts is found in the manuscript on the following lines:*
*ALT size cut information: page 5, lines7-8*
*BRW size cut information: page 5, lines 17-20*
*PAL size cut information: page 5, lines 30-31*
*SUM size cut information: page 6, lines8-9*
*TIK size cut information: page 6, lines19-21*
*ZEP size cut information: page 6, line 31*

*The size cut in Table 1 for PAL was mistakenly reported as 2.5 µm when it should have been 10 µm. This mistake only concerns three sentences in the manuscript. See below: The first sentence is where the size cut is first mentioned along with the description of the station on page 5, lines 30-31. The second sentence on page 14 where lines 17-19 were changed to exclude PAL from the stations with a PM2.5 inlet:*

Page 5, lines 30-31:  SUM statistical values of SAE are not directly comparable to the other Arctic sites due to the 2.5 µm size cut inlet, which limits measurements of large particles that would yield smaller SAE values.

*PAL was removed from the sentence and the reasoning from the discussion about light scattering coefficients was added to the different behavior of the SAE values at PAL:*

Page 17, lines 6-8: PAL and SUM do not show substantial systematic variability in these optical parameters, likely due to their 2.5 µm size cut inlet ( SUM) and/or remote high elevation location (SUM) that limits the measurement of larger particles and thus yields consistently high

SAE values. The different behavior of PAL is likely due to the location of the site (lowest latitude) and difference in the vegetation surrounding the station as discussed earlier.

**Figure captions. I realize that the size range of the measurements is given in the text but it would be useful to provide it in the figure captions, too, since many readers may only look at the figures.**

*We have added size cut information to the captions of Figures 3 and 4, which show seasonality of scattering coefficients and absorption coefficients, respectively.*

Page 34, lines 5-6: "Figure 3. Seasonality of aerosol light scattering coefficient ($\sigma_{sp}$) at 550nm at all sites. Large plot shows monthly medians of scattering in $Mm^{-1}$ at each station, subplots below show boxplots of hourly average scattering at individual sites with horizontal line at the median, edges of the box at $25^{th}$ and $75^{th}$ percentiles, and whiskers at $5^{th}$ and $95^{th}$ percentiles. Note that y-axes are different on each plot. Size cuts for the scattering measurements are as follows: 10 µm (ALT), 10 µm (BRW), 10 µm (PAL), 2.5µm (SUM), 10µm (TIK) and no size cut at ZEP."

Page 35, lines 5-6: "Figure 4. Seasonality of aerosol light absorption coefficient ($\sigma_{ap}$) at 550nm at all sites. Large plot shows monthly medians of absorption in $Mm^{-1}$ at each station, subplots below show boxplots of hourly average absorption at individual sites with horizontal line at the median, edges of the box at $25^{th}$ and $75^{th}$ percentiles, and whiskers at $5^{th}$ and $95^{th}$ percentiles. Note that y-axes are different on each plot. Size cuts for the Aethalometer absorption measurements are as follows: 10 µm (ALT), 10µm (SUM), 10µm (TIK) and no size cut at BRW, PAL, and ZEP."

**Page 12, lines 12 - 14: What are the April and August peaks in light scattering at Summit due to?**

*The goal of the analysis presented here is to document seasonality in Arctic aerosol properties and highlight the differences in aerosol property seasonality between stations across the Arctic, and so the analysis performed here does not directly inform why the aerosol properties have this seasonality. Future work is needed to better understand mechanisms driving the differences in seasonality. However, we can draw from previous studies to help shed light on this question. For one, Stohl et al. (2006) show an instance of forest fires in 2004 impacting absorbing aerosols at SUM, and one could speculate that forest fire season impacts light scattering at SUM as well.*

**Page 12, line 6: It is not clear from Figure 4 that PAL "has the highest absorption coefficients during the summer."**

*This was perhaps a mistake in phrasing. We meant PAL "has the highest absorption coefficients* **of the six Arctic stations** *during the summer".*

*During the months of June, July, August, and September, the highest absorption coefficients are measured at PAL (the green line on the plot). Average Pallas absorption coefficients are at least double the average absorption coefficients measured at the other five Arctic stations. This has been clarified in the text.*

Page 13, line 6: "…, while PAL has the highest absorption coefficients compared to the other Arctic stations during the summer".

**Page 13, lines 28 - 32: Could lower SSA at SUM during September be due to forest fires?**

*Without forest fire data combined with back trajectories, it is impossible to say with certainty if low SSA at SUM in September is due to forest fires specifically. Attribution of elevated aerosol measurements is outside the scope of this paper. However, we know that this explanation is, in theory, possible, as Stohl et al. (2006) show EBC enhancement at SUM during high forest fire activity in July and August of 2004. The relationship of forest fires to seasonality of aerosol optical properties at SUM might be worth further investigation in a separate analysis.*

**Figure 9. It would be helpful to have location markers for the stations on the maps.**

*Thank you for this suggestion. Location markers for the stations have been added to the maps in Figure 9.*

**References:**

Stohl, A., Andrews, E., Burkhart, J. F., Forster, C., Herber, A., Hoch, S. W., Kowal, D., Lunder, C., Mefford, T., Ogren, J. A., Sharma, S., Spichtinger, N., Stebel, K., Stone, R., Ström, J., Tørseth, K., Wehrli, C., and Yttri, K.E.: Pan-Arctic enhancements of light absorbing aerosol concentrations due to North American boreal forest fires during summer 2004, J. Geophys. Res., 111, D22214, doi:10.1029/2006JD007216, 2006.

---

## Author Comment (AC2) · 10 May 2018

**Response to Anonymous Referee #2**

Thank you, anonymous referee #2, for your insightful comments on the manuscript. The paper is more robust thanks to your input.

Our response is structured as follows: original comments from reviewer #2 are bolded, our responses are in italics, and the revised portions of the manuscript follow in quotation marks with specific changes/additions in red.

**Comments**

**In the abstract and in the introduction the authors comment about the importance of studying the in-situ surface aerosol optical properties in the Arctic given the sensitivity of the Arctic climate to short-lived climate forcers.**

**In general, more speculations about the reasons explaining the observed differences among the stations are needed to improve the scientific significance of the presented work. In most cases the manuscript presents a list of extensive and intensive values/properties at each station but the reasons behind the observed aerosol optical properties is sometimes missing.**

> *The authors respectfully disagree with this general comment, as we do not believe speculation improves the scientific significance of a paper. While speculation may guide research questions for future work, the scientific significance lies in the evidence-based aspects of this analysis. Much of the scientific value of this work lies in the finding that aerosol optical properties vary widely with season at each of the Arctic sites, and vary widely from station to station. Though this analysis alone can't explain this spatio-temporal variability, it is a robust springboard for future work exploring the reasons for this.*

**The manuscript should be more focused on the Arctic haze phenomenon. For this, a reorganization of the manuscript is needed. Some suggestions are given below.**

> *While the authors agree that there could be more discussion in the manuscript about the Arctic haze phenomenon in relation to the seasonality of optical properties presented here, we do not think a reorganization is necessary. The goal of this paper was not to focus on the Arctic haze phenomenon, but to present seasonality of aerosol optical properties throughout the entire year. Part of exploring the seasonality means comparing aerosols during the Arctic haze season to aerosols during other parts of the year; consequently, this comment has been incorporated and care has been taken to more specifically address the Arctic haze phenomenon throughout the manuscript, but a restructuring to analyze AH and non-AH time periods separately has not been done. See comments below.*

**1) The six stations included in this work have two types of filter based absorption instruments: the "reference" instrument (CLAP, PSAP, or MAAP) and the Aethalometer model AE31. The AE31 attenuation data are corrected with the Arctic specific correction factor from Backman et al. (2017). The same Cf ( = 3.20) is used to correct the AE31 data from the six observatories. Absorption data collected with the AE31 at 550 nm are presented in this manuscript. The absorption at 550 nm is calculated using the absorption Angstrom exponent (AAE) calculated from the 7-λ Aethalometer measurements. The authors show that the comparison between absorption from the "reference" instrument and the AE31 is "imperfect and variable among stations".**

**Why not present the absorption measurements from the "reference" instruments rescaled to 550 nm using the AAE from the AE31 instruments? If CLAP, PSAP or MAAP are considered as "reference" instruments, then data from these instruments should be presented in the manuscript. Moreover, the same $C_f$ is applied to the seven absorption measurements from the AE31 instruments thus meaning that the AAE from uncorrected AE31 data and the AAE from corrected data should be approximately the same.**

> *The reference absorption instruments at the stations were used to formulate an average best-guess correction factor for Arctic Aethalometers so that data from the same instrument could be used across the stations. Using homogeneously corrected Aethalometer data at all sites has the advantage of the data being more inter-comparable. From Backman et al. (2017): "The benefit of having the same type and model of instrument is that measurement artefacts for the same type of instrument would be expected to be more similar than between different types of instruments. The comparison of aerosol properties between different sites should be more robust when all sites have the same type of instrument than if the instruments would differ from site to site... For the sake of inter-comparability, a relative normalization factor is introduced to harmonize the determination of the absorption coefficient at the Arctic stations."*

**2) Alternatively, if the authors think that the AE31 data are sufficiently robust to be presented in the manuscript (note that the supplemental material is not provided by the authors, so it is difficult to evaluate the goodness of the corrected AE31 data (and consequently SSA values)), then they should take more advantage of the multiwavelength absorption measurements from AE31 instruments.**

**Is there any specific/interesting feature in the AAE seasonality at the six stations? Why not study the spectral dependence of the single scattering albedo (SSA)? For example, presenting the SSA not only in the green, but also in the UV and near IR? The variability of these two quantities (AAE at least) should be discussed in the manuscript.**

> *Our preference is to use Aethalometer data, as we feel these measurements are most robust and inter-comparable between stations. As for studying SSA in the UV and the near IR, this is not possible as our aerosol scattering coefficient measurements from the nephelometers are not made in the UV and the near IR.*

> *AAE values were indeed calculated for each site, and the data were originally included in the manuscript; however, there was not necessarily a specific or interesting feature in the AAE seasonality that was worth including. In order to shorten length of the manuscript, the AAE analysis was cut from the final version. There was little coherent seasonal signal, and differences between most sites was small (a spread of AAE values between 0.7 and 1.1 is actually quite small). For these reasons, the AAE analysis was not included in the manuscript. Included below, however, is the AAE seasonality plot and original description to further motivate this response.*

[Figure]

*From a previous version of the manuscript in which AAE was included: "Absorption Ångström Exponent (AAE) climatologies have not previously been reported for stations in the Arctic. Statistics of AAE at the 520/660 nm wavelength pair, calculated using corrected Aethalometer absorption coefficients, are presented in Table 2. AAE values are not available at SUM, since SUM only has measurements from a 1 wavelength Aethalometer AE16. Two of the Arctic stations, PAL and TIK, have notable seasonality in AAE values. PAL has highest AAE values in the spring, and lowest AAE values in the fall. TIK, on the other hand, has lower AAE values in the spring and early summer, and higher values of AAE in the fall. These changes in AAE statistics throughout the year suggest that these sites might measure different aerosol compositions depending on the season; however, the range in AAE values is fairly minimal."*

**3) In the manuscript the Arctic Haze (AH) phenomenon is discussed together with the scattering and absorption measurements. ALT, BRW, TIK, and ZEP stations present an increase in both scattering and absorption in late winter/spring related to the AH phenomenon. However, there is no mention to the AH phenomenon in the sections presenting the intensive aerosol optical properties.**

**a) A table presenting the mean SAE, SSA, g (and possibly AAE) during AH period versus non-AH period should be presented and discussed. The spatial differences (from one site to another) in the intensive optical properties during AH period should be also discussed. For example, the seasonality of scattering and absorption at ALT, BRW, TIK and ZEP is very similar (and ascribed by the authors to the AH phenomenon) whereas the intensive properties are very different. For example, the SSA at ALT during AH is much higher (and different in term of seasonality) from the SSA observed at TIK during the AH period. The authors should comment/discuss the possible reasons explaining why the intensive properties change from one site to another during AH period.**

*We agree that more discussion of differences between intensive aerosol optical properties during the Arctic Haze season and the rest of the year is needed, and additions have been made to the manuscript. No additional table has been added to document the different*

*statistics between the Arctic Haze and non-Arctic Haze seasons, as the authors feel the differences between these time periods is easy to see on the monthly climatology plots in Fig. 3-7, and additionally, the focus of the manuscript is not to differentiate specifically between the AH and non-AH seasons but to document the seasonality throughout the entire year.*

**b) It is interesting the fact that the effect of AH on intensive properties is not observed at PAL and SUM which are located at higher altitude compared to the other stations. Is there any relationship between altitude of the station and AH phenomenon?**

*You are right that the fact that PAL and SUM do not show a large Arctic Haze season signal in their intensive aerosol optical properties is interesting; it might be worthwhile to dig deeper into the relationships between altitude of a measurement station and the AH phenomenon. However, there are not enough high elevation vs. low elevation observatories to say with much certainty whether or not differences between stations is due to elevation alone or other environmental differences. This may be a question more appropriately answered with a modeling study.*

**4) The authors say that "..surface Arctic aerosol optical properties in particular can help define and constrain inter-annual, seasonal and diurnal variability" (Pag. 2, Line 22-23). Why not present the diurnal cycles of both extensive and intensive aerosol particle optical properties? This can be done comparing AH period versus non-AH period.**

*We did perform a short analysis of diurnal variability during the preparation of the manuscript, though we felt the results were not worth including in this paper in particular. Since diurnal variability is not large at most stations, there was not much to comment on with regards to station to station differences in diurnal variability. We have addressed your comment by removing "diurnal variability" from page 22, line 22-23. See below for plots of hourly climatologies of aerosol absorption and scattering coefficients.*

[Figure]

**5) Improve the abstract. In the present form the abstract present a list of lowest/highest values of extensive and intensive properties at the six observatories, but the reasons/speculations behind the variability of the reported values is missing.**

*The authors do not feel comfortable including any speculations in the abstract, as we do not directly in this analysis provide evidence that explains most of the seasonality. The important finding of this paper is in the spatio-temporal variability of surface aerosol optical properties in the Arctic.*

**6) Pag. 8, Line 31. Figure 2 shows the time series of monthly median corrected AE31 data. Why not present the daily median? Note also that the supplemental material was not uploaded. Consequently, it is very difficult to evaluate the goodness of the comparisons using just monthly medians.**

*Our apologies that the supplemental material was not uploaded, it was made available on the ACPD portal soon after you noticed this. Comparing daily medians from the AE31 made for a noisy and difficult-to-read plot, whereas the plot of monthly medians conveyed the overall comparison in a more concise way, without losing the essential information. Furthermore, monthly medians have the distinct benefit of a much larger signal to noise ratio compared to daily medians, which as you point out in later comments, is especially important when making measurements in clean Arctic conditions.*

**7) Pag. 9, Line 33: The authors should explain where the data came from. For example, was it downloaded from EBAS. Or was it provided by data providers?**

*In the Data Availability section at the end of the manuscript, it is stated that all data used in the article are archived and accessible from the EBAS database. This sentence has also been added to the manuscript.*

Page 9, line 33: "All data used in this analysis are archived and accessible from the EBAS database operated by the Norwegian Institute for Air Research (NILU)."

**8) Pag. 10, Line 8: Add that also g was one of the variables considered in the manuscript.**

*Thank you for catching this. We have added g to the list of variables in this paragraph.*

Page 10, lines7-8: The variables analyzed here include extensive aerosol optical properties that depend on aerosol amount, absorption ($\sigma_{ap}$) and scattering ($\sigma_{sp}$) coefficients and asymmetry parameter (g), ...

**9) Pag. 10. How were the intensive properties calculated? Using all the scattering and absorption data or using only data above a given threshold (i.e. >1 Mm-1)?. Calculating the intensive properties using scattering or absorption data higher than a given threshold is important in order to remove undesired noise in the calculations.**

*Thank you for pointing this out. Given the incredibly small absorption and scattering coefficients measured at the Arctic sites, the typical thresholds of, say, >0.5 Mm$^{-1}$ for absorption coefficients and >1 Mm$^{-1}$ for scattering coefficients are not used in this analysis because it would eliminate a very large portion of the data measured at these sites. During some months (e.g., summer months when very low scattering coefficients are measured), that would exclude nearly all of the data.*

*Given the clean conditions in the Arctic, removing undesired noise in the data was accomplished through temporal averaging. The authors understand that computing intensive aerosol optical properties with low values of extensive aerosol optical properties can be problematic, and thus that is one reason why monthly medians are presented throughout the paper, as the authors have more confidence in a larger signal-to-noise ratio of monthly values compared to hourly or daily values. Furthermore, it was noted throughout the paper (page 14 line 31- page 15 line 1) that when intensive properties show large variability it is likely in part due to some noise from low scattering and/or absorption measurements.*

**For example: In Figure 2 the SSA at ALT and SUM in July and September, respectively, presents the lowest values when also scattering and absorption are low. The same is observed for the scattering Angstrom exponent at SUM in winter or the asymmetry parameter at SUM in January (for example).**

*Furthermore, non-physical values of SSA (i.e., SSA<0 or SSA>1), for example, were removed from computations of monthly and annual statistics.*

**How do these figures (Figures 5, 6, 7) change if a threshold is applied before calculating the intensive properties? In the case of SSA at SUM in September the authors speculate that the low SSA is related to an increase in flights and transportation activity. However, for other stations/seasons no explanations are given to justify why the 5th and 95th percentiles are too low or high. It is important to demonstrate that these high deviations of intensive properties at some stations are not due to noise.**

*Figures 5, 6, and 7 would not necessarily be representative of typical Arctic conditions if a threshold was applied before calculating the intensive properties, as typical thresholds used in this type of aerosol optical property analysis (>0.5 $Mm^{-1}$ for absorption coefficients and >1 $Mm^{-1}$ for scattering coefficients), since median values of extensive aerosol optical properties at many of these Arctic stations are much below these thresholds. The authors discussed this problem in preparation of the manuscript, and decided it would not be prudent to remove data below these thresholds as it would eliminate so much of the dataset. Instead, noise was reduced by temporal averaging to monthly medians, then care was taken to remove intensive aerosol optical property values that were non-physical, for example, SSA values below 0 or above 1. The number of intensive aerosol optical property values removed with this 'threshold' was small.*

*For the purposes of considering this reviewer comment further, the authors administered slightly less strict thresholds (>0.25 $Mm^{-1}$ for absorption coefficients and >0.5 $Mm^{-1}$ for scattering coefficients) and remade the figures to see how they would change. With this threshold, much of the data points were removed. For ALT, this left only 31% of the original data points. The percentage of data points remaining after applying the thresholds at other stations was as follows: 19% (BRW), 24% (PAL), 4% (SUM), 9% (TIK) and 15% (ZEP).*

*Aerosol absorption coefficient medians with thresholds (see figure below), for example, do not look very different from those medians without thresholds. This finding is robust for Figures 4-7; though scattering medians do get higher when filtering out all scattering measurements below 0.5 $Mm^{-1}$.*

[Figure]

*Single scattering albedo medians computed after scattering and absorption coefficient thresholds are applied are shown in the figure below. At all stations with the exception of SUM, there is little change compared to medians without thresholds. The exception is SUM, which shows much lower SSA values than without the thresholds applied, particularly for the months of March, September, and October. Bear in mind that with the absorption and scattering thresholds applied, only 4% of the data points at SUM remained. In some months (January and November), this means there is no intensive data left at SUM at all. In the months that do have intensive optical property data available, the values are highly skewed towards what could be considered as 'polluted' events at this otherwise very clean site. This arguably skews SSA values to much lower than they 'normally' are at SUM, and therefore we argue that the seasonality shown below, when thresholds are applied, is not representative of typical conditions at these sites, especially SUM.*

[Figure]

**10) Pag. 10, Equation 2: Why not present the differences between the SAE calculated between 450 and 550 nm and the SAE calculated between 550 and 700 nm? Is there any interesting difference between the two SAE during the AH phenomenon versus periods without AH phenomenon?**

*This is an interesting question that could be worth exploring in another analysis, but due to time constraints, will not be investigated here.*

**11) Pag. 10. The AAE from AE31 was used to calculate the absorption at the same wavelength of the "reference" instrument. How was the AAE calculated? Were used all the wavelengths or only those close to the reference wavelength?**

*Thank you for asking us to clarify this. AAE was calculated with the following equation:*

$$AAE = -\frac{\log(\sigma_{a1}) - \log(\sigma_{a2})}{\log(\lambda_1) - \log(\lambda_2)}$$

*All of the wavelengths were not used. For this analysis, AAE values were calculated with the 520 nm and 660nm wavelength pair, the pair closes to the reference wavelength of 550 nm. This detail has been added to the manuscript.*

Page 11, lines 9-10: "Absorption measurements were adjusted to the same wavelength with the AAE value calculated from the 520 nm/660 nm wavelength pair."

**Moreover, (end of Pag. 10 – beginning of Pag. 11), the authors say that the SAE was also used for the wavelength adjustment of nephelometer data. However, the TSI nephelometer works at 550 nm which is the wavelength used to present the results. So, no adjustment of nephelometer data is in principle needed. Please, clarify.**

*Thank you for catching this. You are right, only data at 550 nm was in the end presented in the manuscript, therefore we have removed this sentence from the text.*

**12) Pag. 13, Line 6. PAL -> SUM**

*We still mean to say PAL here, but have added a clarifying term to indicate that we mean PAL has the highest absorption coefficients in the summer compared to the other Arctic stations.*

Page 13, line 6: …PAL has the highest absorption coefficients during the summer compared to the other stations.

**13) Pag. 13, Lines 16-18: Explain why at ALT the SSA values drop during July (any physical explanation or noise?)**

*Upon further investigation of the ALT data in July, it is clear that there are many days of very low scattering measurements in that month that contribute to the low SSA values. This analysis alone cannot provide certainty about a physical explanation for why SSA could be so low. However, as mentioned on page 16 in lines 22-24 which comments on the systematic variability in Figure 8: "The SSA vs. scattering relationship here suggests that whiter aerosols are preferentially scavenged such that darker aerosol remain at the lowest aerosol loadings (lowest scattering coefficients)".*

*For the purposes of addressing the reviewer comments, the threshold of $\sigma_{ap} > 0.25$ Mm$^{-1}$ and $\sigma_{sp} > 0.5$ Mm$^{-1}$ were applied to the ALT data. When the thresholds are applied, only 8,379 observations out of the total 26,304 aerosol observations made at ALT remain (only 89 observations in July). SSA was then calculated using only measurements that were greater than the thresholds, and values of SSA from filtered and non-filtered data were compared during July at ALT. For July in ALT with no thresholds applied to the data, the median SSA=0.90, while data with thresholds applied give a median SSA=0.87. The median of SSA value is slightly less with thresholds applied compared to data without thresholds.*

**14) Pag. 13, Line 18: Explain why the SSA values at BRW are the highest during September-October.**

*The high SSA average at BRW during this time is likely in part due to the minimum sea ice extent and thus increased open ocean and sea salt aerosol during September and October. Figure 10 lends some evidence towards this. Furthermore, at least one other publication supports this hypothesis (May et al., 2016), and thus a sentence speculating on this has been added to the manuscript.*

Page 13, line 22-24: "SSA values at BRW could be highest in September and October due to low sea ice extent, more open ocean and thus the potential for more sea salt aerosol in the area (May et al., 2016). Figure 10 lends evidence for this, and is discussed later in the manuscript."

**15) Pag. 13, Lines 22-24 ("This is explained by ……….. is low and scattering is high"). Remove the sentence. This is obvious.**

*This sentence has been removed.*

**16) Pag. 13, Line 25 and Lines 27-28: The high scattering at PAL in summer is probably due to the enhanced formation of BSOA. This is probably consistent with the fact that absorption does not show the same increase in summer. Consequently, the SSA is the highest in summer (with quite low standard deviation of the data) and reflects the presence of very "white" particles. However, the authors say (Line 25) that there is an increased contribution from continental air masses in the summer at PAL. So, what is driving the evolution of the extensive and intensive properties at PAL in summer? The arriving of continental air masses (probably containing less "white" particles) or the BSOA formation (Lines 27-28)?**

*The seasonality of aerosol optical properties at PAL is very likely a combination of multiple factors, including an increase in continental air masses arriving at the station, a decrease in anthropogenic sources like wood burning (given higher summer temperatures, residential heating in Europe is no longer needed), and an increase in biogenic secondary organic aerosol formation. Chemical measurements and smaller scale trajectory analyses are needed to fully answer this question. This could be another manuscript in itself, and is likely worth exploring!*

**17) Pag. 14, Lines 24: Also here it is important to demonstrate that the large variability in SAE in July-September at TIK (when scattering and absorption are very low) is not due to noise. It is important to know if any threshold has been applied before calculating the intensive properties.**

*See also the answer to reviewer comment #13. In the manuscript, no thresholds were applied to extensive properties before calculating the intensive properties because this would eliminate so much of the available data. However, for the purposes of addressing the reviewer comments, the threshold of $\sigma_{ap} > 0.25$ $Mm^{-1}$ and $\sigma_{sp} > 0.5$ $Mm^{-1}$ were applied to the TIK data. When the thresholds are applied, only 2,444 observations out of the total 26,304 aerosol observations made at TIK remain. Intensive properties were then calculated using only data that were greater than the thresholds, and values of SAE, for example, were compared to filtered and non-filtered data during summer months at TIK. For July in TIK with no thresholds applied to the data, the median $SAE_{450/700nm} = 1.71$, while data with thresholds applied give a median $SAE_{450/700nm} = 1.74$. The medians of SAE values are nearly the same with or without thresholds, and this finding is robust across summer months at TIK and across intensive aerosol optical properties.*

*It is also worth mentioning that it has been acknowledged elsewhere in the paper (page 14 line 36 – page 15 line 1) that large variability in intensive properties tends to be concurrent with periods of low scattering and/or absorption measurements.*

**18) Pag. 14, Line 32: It seems that g also varies quite a lot from one station to another, whereas the authors say that "the asymmetry parameter, g, is similar for all sites except for SUM". Please, clarify/expand.**

*The authors stated that "the variability of the asymmetry parameter, g, is similar for all site except for SUM". This was meant to communicate that all stations except for SUM show the same general seasonality with larger values of g in the winter and smaller values of g in the summer.*

**19) Pag. 15, Line 7 and Figure 8: Why not show the g too?**

*Systematic variability of g with the other aerosol optical properties was explored (see one of the plots below), though nothing of particular interest was found in those plots, and thus they were not included in the manuscript.*

[Figure]

**20) Table 2: SUM station registers the highest SAE (small particles) and also the highest g (large particles). Any explanation for this?**

*The authors agree this is a bit of an enigma, but since g also depends on shape and composition in addition to size, this suggests shape and composition are likely playing a role here. We cannot with certainty from the analysis here what could explain this. Aerosol size distribution measurements and size-segregated chemical composition measurements could help with answering this question, but neither is available at SUM.*

**21) Section 4.3: Figure 10 is nice. It seems that there is a relationship between the time spent above open water and sea ice and Figures 3 and 4. For example, at TIK the scattering is the lowest when air masses spent more time over sea ice and open water (June to September). At ZEP the reduction of scattering between June and October reflects the relative increase of time spent over sea ice and open water (and less time spent over land). Can the authors say something more about Figure 10? Is it possible to relate the time spent over land with the Arctic haze phenomenon? The paragraph at Pag. 18, Lines 16-29 should be expanded.**

*We have taken your comments into consideration and have expanded the discussion throughout the paper to include more results from Figure 10.*

**Figure 9 seems less useful. The highest frequency is always observed for regions close to the stations. Why not use, i.e., the potential source contribution function or the concentration weighted trajectory? (Both are available for example in the OPENAIR r package). These plots could be colored by levels of scattering and absorption to get a clearer idea about source regions. The differentiation in terms of air masses between AH periods versus non-AH periods should be introduced and discussed.**

> *We would argue that Figure 9 is useful in that it shows highest frequency is mostly symmetric around the station locations. For most stations, a preferential trajectory path does not jump out in either summer or winter. This result, although somewhat boring, is still valuable to know.*
>
> *Your suggestion about concentration weighted trajectories is a good one, and concentration weighted trajectories were indeed already plotted during the process of this analysis. They showed interesting results, and are already being included in a forthcoming publication in preparation. Thus, those results are not shown here.*

New References:
May, N. W., P. K. Quinn, S. M. McNamara, and K. A. Pratt (2016), Multiyear study of the dependence of sea salt aerosol on wind speed and sea ice conditions in the coastal Arctic, J. Geophys. Res. Doi.org/2016JD025273.

---

## Author Response (AR2)

**Authors' Response to Reviewer Comments on
'Seasonality of Aerosol Optical Properties in the Arctic'**

**Response to Anonymous Referee #2**

Thank you, anonymous referee #2, for the additional comments. Our response is structured as follows: additional comments from reviewer are bolded, our responses are in italics, and the revised portions of the manuscript follow in quotation marks with specific changes/additions in red.

**Comments**
**I thank the authors for their responses that address the majority of my comments.**
**However, I think that some points could be better addressed/commented. My additional comments below.**

**Comment#2- AAE: In your first version of the manuscript you commented this "lack of interest" of presenting AAE. Consequently, you decided to remove from the text that comment. However, I think that at least a couple of sentences on AAE should be presented in the paper and the Figure presented in the Supplement Material. As you have highlighted in your response, the climatology of AAE in the Arctic has never been presented before. Moreover, I would not use the 520/660 nm wavelength pair to calculate the AAE. Rather, I would use the entire measured spectrum calculating the AAE as fit of the 7 wavelengths.**

*The authors computed and plotted AAE spanning all 7 Aethalometer AE31 wavelengths as requested by the reviewer. The difference in seasonality between AAE at the 520/660nm wavelength pair and the 370/950nm wavelength pair is small, as shown in the figures below. The most notable difference in $AAE_{520/660nm}$ and $AAE_{370/950nm}$ seasonality is during the winter at PAL and TIK where $AAE_{370/950nm}$ is larger than $AAE_{520/660nm}$.*

[Figure]

[Figure]

*A paragraph describing our AAE methods and the resulting seasonality of AAE as well as the figure of AAE seasonality has been added to the supplemental materials.*

In supplemental materials: "Absorption Ångström exponent represents the wavelength dependence of the aerosol absorption coefficient. This intensive aerosol optical property depends on aerosol composition, so different aerosol types have unique ranges of AAE values. AAE is computed using the following equation:

$$AAE = -\frac{\log(\sigma_{a1}) - \log(\sigma_{a2})}{\log(\lambda_1) - \log(\lambda_2)} \qquad (S1)$$

where $\sigma_{a1}$ is the light absorption coefficient at wavelength $\lambda_1$, and $\sigma_{a2}$ is the light absorption coefficient at the wavelength $\lambda_2$.

Figure S2 shows seasonality of absorption Ångström exponent (AAE) at 5 sites in the Arctic. AAE climatologies have not previously been reported for stations in the Arctic. AAE values are not available at SUM, since SUM only has measurements from a 1 wavelength Aethalometer AE16. Two of the Arctic stations, PAL and TIK, have notable seasonality in AAE values. PAL has highest AAE values in the winter and spring, and lowest AAE values in the summer and fall. TIK, on the other hand, has lower AAE values in the spring and early summer, and higher values of AAE in the fall. These changes in AAE statistics throughout the year suggest that these sites might measure different aerosol compositions depending on the season; however, the range in AAE values is fairly minimal."

In main manuscript, page 11, line 7-8: "For those further interested, the seasonality of absorption Ångström exponent is presented in supplemental materials (Figure S2)."

**Comment#2- SSA: Undoubtedly, the nephelometer does not measure in the UV and in the near-IR. Despite this, the spectral dependence of SSA from UV to near-IR has been already published assuming that the SAE calculated in the visible can be used to calculate the scattering from UV (370nm) to near-IR (950nm). I do not know if the high S/N of the calculated UV and near-IR scattering and of the measured absorption can mask any interesting feature. It is impossible to know this without performing the calculations. I would like at least to know if there is any**

**interesting feature. If yes, maybe a couple of sentences could be added to the manuscript.**

*After extrapolating nephelometer scattering coefficients to 370nm and 950nm, SSA values at 370nm and 950nm were then calculated. Below are plots of SSA at 370nm, 550nm, and 950nm for each of the stations (except SUM since absorption measurements are not available at those wavelengths). The plots can be interpreted as follows: when $SSA_{950nm}$ is greater than $SSA_{370nm}$, it generally indicates the domination of coarse particles like dust or sea salt; conversely, when $SSA_{370nm}$ is greater than $SSA_{950nm,}$ as is the case most of the time at these Arctic stations, it indicates domination by smaller particles from smoke and biomass burning (Costabile et al., 2013; Bergstrom et al. 2002). This interpretation aligns with what we see in the SAE seasonality plot (Figure 6 in the manuscript, shown again here in the bottom right of the table). Take, for example, BRW during August, September and October, when $SSA_{950nm}$ exceeds $SSA_{370nm}$, indicating presence of coarse particles. SAE during those months drops to a minimum- below 0.5- also indicating the presence of coarse particles. Another example where the SSA dependence aligns with the SAE seasonality is at PAL during March and November, when $SSA_{950nm}$ exceeds $SSA_{370nm}$ and SAE also drops during those months. Since the SSA wavelength dependence does not add any additional information beyond what we get from the SAE, and since the authors do not feel comfortable presenting scattering data extrapolated that far out of the nephelometer range, nothing new is added to the manuscript.*

[Figure]

**Comment#10: As in my comment above. I do not know how much the high S/N affects these calculations. Moreover, the Arctic is a very pristine environment and calculating SAE pairs could not add any interesting information. But, I think you could try to calculate these SAE pairs and, in case something interesting come up, a short paragraph could be added to the manuscript.**

*As suggested by the reviewer, SAE$_{450/550nm}$ and SAE$_{550/700nm}$ seasonality were compared, as shown in the figures below. The figures show that the SAE for the different wavelength pairs exhibit very similar seasonality although there are some differences in SAE values at the different wavelength pairs. An analysis of these differences is beyond the scope of this paper but could be the subject of an additional study, particularly for any stations where the full aerosol size distribution (i.e., from measurements by a SMPS+APS) was available. The dataset used in our analysis will be publically available, and readers interested in analyzing the differences between SAE values at different wavelength pairs will be able to do so.*

[Figure]

**Comment#19 and #20: I think the scat vs. g Figure you report in your response is quite interesting. It confirms what you mention in your response #20 that shape and size distribution affect differently g and SAE. In the scat vs. g figure g increases with scattering at all stations (except at PAL) indicating a progressive shift of the size distribution toward large particles maybe in the**

**accumulation mode (consistent with aging for example). The increase of g with increasing scattering is consistent with the progressive decrease of SAE at BRW. The slight decrease of g with increasing scattering at PAL seems to be consistent with the increase of SAE. At all other stations it seems that a clear relationship is not observed. Look for example at ZEP where the SAE is almost constant whereas the g clearly increases. This is very probably related to the aerosol size distribution. I agree to not present the size distributions in this paper, but I would add the scattering vs g Figure in the manuscript.**

*As suggested by the reviewer, the figure of scattering coefficient vs. asymmetry parameter has been added to the supplemental materials, along with a description. We also added a short reference to the figure in the main manuscript in the section discussing Figure 8c.*

In the supplemental materials: "Figure S3 shows the systematic variability of median aerosol asymmetry parameter (g) varying with scattering coefficient ($\sigma_{SP}$) at all of the Arctic sites. At all stations (with the exception of PAL) there are increasing g values with increasing $\sigma_{SP}$. This generally means that as scattering increases, particles tend to be larger. Conversely, the cleanest days when scattering is lowest tend to be dominated by smaller particles. The increasing g with increasing $\sigma_{SP}$ is consistent with aging. Furthermore, the observed systematic variability is consistent with the decrease in SAE with increase in $\sigma_{SP}$ at BRW, though this is not the case at other stations. This further confirms the idea presented in the main manuscript that the specific shape of the aerosol size distribution at each site is important in determining the unique seasonality of g and SAE at each of the Arctic stations, since each of these parameters depends more strongly on different parts of the aerosol size distribution. At PAL, where g does not change much with $\sigma_{SP}$, or even decreases slightly with increasing $\sigma_{SP}$, it is consistent with the observed increase in SAE with $\sigma_{SP}$."

In the main manuscript, page 17, lines 3-5: "The asymmetry parameter provides another means of investigating changes in particle size distribution with loading, and the plot of scattering vs. asymmetry parameter and the associated discussion are included in the supplemental materials (Figure S3)."

[revised manuscript text omitted]